# Enhanced Photocatalytic Oxidation of RhB and MB Using Plasmonic Performance of Ag Deposited on $Bi_2WO_6$

**Shomaila Khanam** and **Sanjeeb Kumar Rout** *

Department of Physics, Birla Institute of Technology, Mesra 835215, India; shomaila27t@gmail.com
* Correspondence: skrout@bitmesra.ac.in

**Abstract:** Visible-light-driven heterostructure $Ag/Bi_2WO_6$ nanocomposites were prepared using a hydrothermal method followed by the photodeposition of Ag on $Bi_2WO_6$. A photocatalyst with a different molar ratio of Ag to $Bi_2WO_6$ (1:1, 1:2 and 2:1) was prepared. The catalytic performance of $Ag/Bi_2WO_6$ towards the photocatalytic oxidation of rhodamine B (RhB) and methylene blue (MB) was explored. Interestingly, the $Ag/Bi_2WO_6$ (1:2) catalyst exhibited superior performance; it oxidized 83% of RhB to Rh-110 and degraded 68% of MB in 90 min. This might be due to the optimum amount of Ag nanoparticles, which supported the rapid generation and transfer of separated charges from $Bi_2WO_6$ to Ag through the Schottky barrier. An excess of Ag on $Bi_2WO_6$ (1:1 and 2:1) blocked the active sites of the reaction and did not produce the desired result. The introduction of Ag on $Bi_2WO_6$ improved the electrical conductivity of the composite and lowered the recombination rate of charge carriers. Our work provides a cost-effective route for constructing high-performance catalysts for the degradation of toxic dyes.

**Keywords:** photodeposition; photocatalytic oxidation; Ag; $Bi_2WO_6$; localized surface plasmon resonance

## 1. Introduction

The discharge of dye effluents in water is a global problem. Different techniques have been employed to treat dye polluted water. Among the variety of methods, photocatalytic degradation using semiconductors as photocatalyst has gained worldwide attention. Researchers around the world are working on improving the efficiency of photocatalysts. Among these, the localized surface plasmon resonance (LSPR) of noble metal/semiconductor heterojunction composite has proven to be much beneficial.

Localized surface plasmon originates in noble metals nanostructures and has very high absorption and scattering cross-section in the region where it occurs. Plasmonic nanostructures develop an oscillation of electrons when incident to the light wave of plasmonic resonance frequency, which produces a bound or localized electromagnetic mode in a confined plasmonic nanostructure. The localized surface plasmon resonance (LSPR) has proven to be highly effective in heterogeneous photocatalysis. Single component photocatalysts have low catalytic efficiency and could not fulfill the desired requirements.

Recently, a number of studies on noble metal-semiconductor hybrid photocatalysts have been performed, and it has proven successful in many reactions, such as degradation, hydrogen evolution, hydrogenation and oxidation [1]. The noble-metal-semiconductor hybrid has found a strong place in the field of photocatalysis. Loading of a noble metal on a photocatalyst can result in an extended light response and can enhance the interfacial charge transfer efficiency [1–5]. The enhanced photoreactivity is credited to localized plasmon resonances of plasmonic nanostructures, in which a metal-semiconductor composite, when illuminated by electromagnetic light of frequency equal to the plasmon resonance frequency of the noble metal of the composite, absorbs the incident photons, giving rise to intense electromagnetic fields [6–9].

This enhanced local electric field leads to an increased interband transition rate making the energy generated by LSPR higher than the bandgap of a semiconductor and increases the electron–hole pair separation in the photocatalyst. The electrons are directed towards a noble metal and holes are accumulated on the other edge of a semiconductor. Noble metal acts as an electron trapper and reduces the recombination rate of electron–hole pairs. This process enhances the photoactivity of the photocatalyst. The enhanced local electric field exists only around the noble metal, and it decays as a function of distance [10]. In contrast, too much noble metal reduces the active sites for the reaction and acts as a recombination center. A high concentration of noble metals blocks the active sites of the reaction.

Figure 1 shows a noble metal on the semiconductor, which absorbs the incident light and undergoes surface plasmon oscillation, which excites electrons and holes. However, the plasmonic enhancement of photoconversion is still a great challenge. To progress in this method, some basic problems, such as the fabrication size, geometry and combination (molar ratio) of a noble metal and semiconductor, need to be thoroughly investigated. The energy transfer between plasmonic metals and semiconductors takes place through three mechanisms: light scattering, hot electron injection and plasmon-induced resonance energy transfer [11–13]. Designing a plasmonic metal-semiconductor photocatalyst is a great challenge. However, some reported work shows that strongly coupled metal-semiconductor nanostructures generated a high intensity of LSPR; however, the kind of architecture of metal-semiconductor heterostructures remains a mystery [14–16].

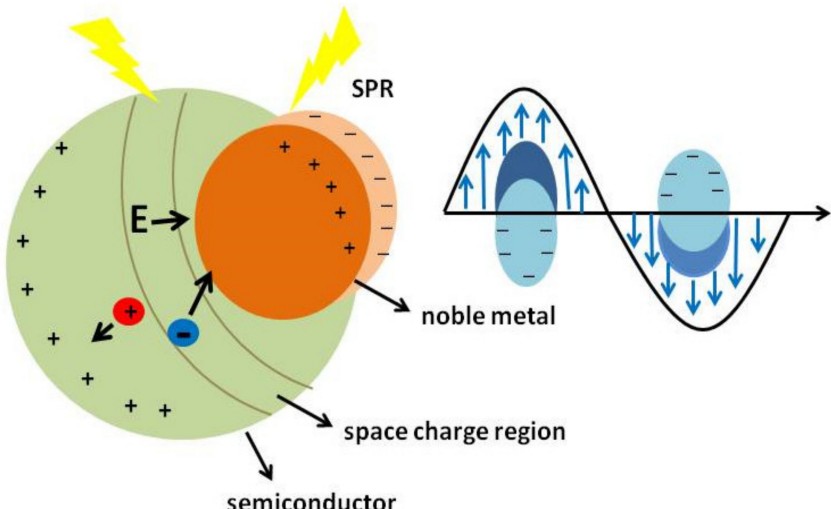

**Figure 1.** Schematic diagram representing charge separation in Ag/Bi$_2$WO$_6$.

There are studies available that investigated metal and semiconductor combinations, such as the fabrication of Ag decorated g-C$_3$N$_4$/LaFeO$_3$ Z-scheme heterojunction by Weijin Zhang and team showed the degradation efficiencies of pollutants MB and TC up to 98.97% in 90 min and 94.93% in 120 min, respectively [17]. Ming Lui reported the improved photocatalytic performance from Bi$_2$WO$_6$@MoS$_2$/graphene hybrids via a gradual charge-transferred pathway and studied the photocatalytic evaluation using RhB degradation [18].

A study reported by Hongwei Tian showed the electron transfer pathway of the ternary TiO$_2$/RGO/Ag nanocomposite with enhanced photocatalytic activity and reported 91% MB degradation under visible light in 130 min [19]. Shuya Xu reported that Ag/Ba-TiO$_3$ showed an excellent photocatalytic property and degraded 83% RhB in 75 min, which is 20% more than the sole photocatalyst TiO$_3$ [20]. Similarly Kaja Spilarewicz modified the surface of TiO$_2$ with Ag and graphene nanostructures and reported the enhanced photocatalytic performance by degrading 93% RhB in 300 min [21]. Yutong Liu and his team reported the excellent photo catalytic performance of Ag modified ZnO and recorded 100% degradation of MB in 40 min [22]. Table 1 summarizes the photocatalytic activity of different photocatalysts.

**Table 1.** Summary of photocatalytic degradation of RhB and MB by different photocatalysts.

| Catalyst | Organic Pollutant | Light Source | Time (Min) | Degradation % | Reference |
|---|---|---|---|---|---|
| $Ag-BaTiO_3$ | RhB | visible | 75 | 83 | [20] |
| $TiO_3$ | RhB | visible | 75 | 63 | [20] |
| Nano-disc $Fe_2O_3$ | RhB | visible | 45 | 13 | [23] |
| $TiO_2$-GO-AgNs | RhB | visible | 300 | 93 | [21] |
| $Ag_2CO_3$-$WO_3$ | RhB | visible | 120 | 64.6 | [24] |
| g-$C_3N_4$ | RhB | visible | 90 | 24 | [25] |
| $WO_3$-g-$C_3N_4$ | RhB | visible | 90 | 62 | [25] |
| $Bi_2WO_6$ | RhB | visible | 10 | 40 | Present work |
| $Ag/Bi_2WO_6$ (1:1) | RhB | visible | 10 | 40 | Present work |
| $Ag/Bi_2WO_6$ (1:2) | RhB | visible | 10 | 50 | Present work |
| $Ag/Bi_2WO_6$ (2:1) | RhB | visible | 10 | 50 | Present work |
| $TiO_2$ | MB | visible | 120 | 40 | [26] |
| $BaIn_2O_4$ | MB | visible | 120 | 53 | [26] |
| $SrIn_2O_4$ | MB | visible | 120 | 67 | [26] |
| $Au/TiO_2$ | MB | visible | 20 | 94 | [27] |
| Ag-ZnO | MB | visible | 40 | 100 | [22] |
| $Bi_2WO_6$ | MB | visible | 90 | 38 | Present work |
| $Ag/Bi_2WO_6$ (1:1) | MB | visible | 90 | 53 | Present work |
| $Ag/Bi_2WO_6$ (1:2) | MB | visible | 90 | 68 | Present work |
| $Ag/Bi_2WO_6$ (2:1) | MB | visible | 90 | 43 | Present work |

In the present study, we aim to produce plasmonic $Ag/Bi_2WO_6$, which can generate a high-intensity electromagnetic field and can prove efficient in degrading RhB and MB. Here, silver is our primary choice due to its high stability and low cost [28,29]. Bismuth tungstate has a narrow bandgap of 2.7 eV and is a visible-light-driven photocatalyst. Moreover, the photocatalytic activity of $Bi_2WO_6$ is limited due to its low absorbance of light and fast photogenerated electron–hole recombination [30,31]. Therefore, surface modification of $Bi_2WO_6$ is required to enhance its photocatalytic efficiency. Forming heterojunction of $Bi_2WO_6$ with silver will increase its light absorbance capacity and the LSPR effect of Ag will reduce its electron–hole pair recombination [28,29].

## 2. Experimental

### 2.1. Synthesis of $Bi_2WO_6$

In a typical hydrothermal procedure, 1.23 g of $Na_2WO_4 \cdot 2H_2O$ and 3.64 g of $Bi(NO_3)_3 \cdot 5H_2O$ were added to a Teflon vessel containing 150 mL deionized water under magnetic stirring. The Teflon vessel was sealed in an autoclave and heated, and the temperature was set to 160 °C for 20 h. After the specified time of the reaction, the autoclave was allowed to cool down naturally to room temperature. The sample was centrifuged and washed several times with deionized water and then dried in an oven at 80 °C for 10 h. Finally, a yellowish nanopowder of $Bi_2WO_6$ was obtained.

### 2.2. Synthesis of Plasmonic $Ag/Bi_2WO_6$

For the photo-induced deposition of Ag on $Bi_2WO_6$, we opted for the introduction of Ag on the semiconductor. In the typical synthesis process, 0.085 g (0.5 mmol) of $AgNO_3$ was added to a beaker containing 50 mL of deionized water and constantly stirred for

30 min in the dark. Then, 0.349 g (0.5 mmol) of hydrothermally prepared $Bi_2WO_6$ was added to the $AgNO_3$ solution and kept in a visible light chamber for 60 min. The grey precipitate was obtained, which was washed thoroughly with deionized water several times and placed in an oven for 8 h at 60 °C. Three different molar ratios (1:1, 1:2 and 2:1) of $Ag/Bi_2WO_6$ were prepared by changing the molar concentration of silver and bismuth tungstate. Figure 2 gives the pictorial representation of the synthesis of $Ag/Bi_2WO_6$.

### Hydrothermal synthesis of nanoceramic

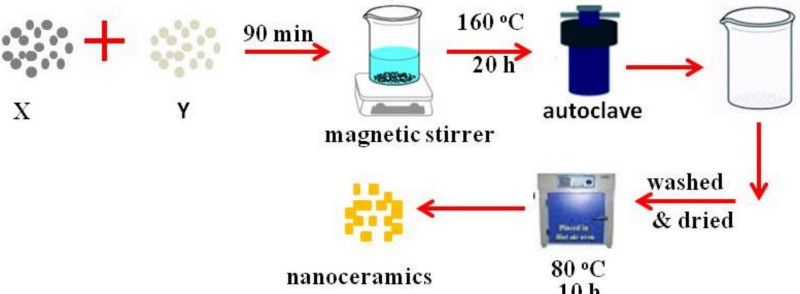

### Photo induced deposition of noble metal on nanoceramic

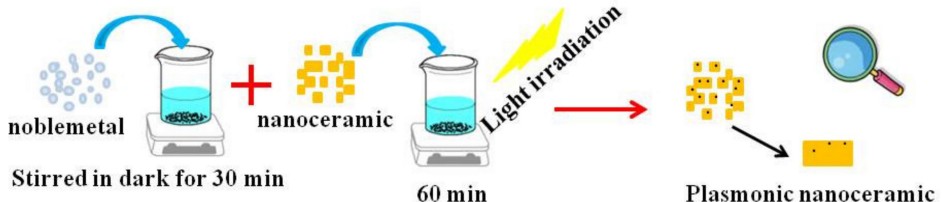

**Figure 2.** Schematic representation of the synthesis of $Ag/Bi_2WO_6$.

### 2.3. Characterization

The phase purity of the prepared photocatalysts was examined by X-ray diffraction (XRD) patterns using (Rigaku, Tokyo, Japan) Smart lab diffractometer. The scanning rate was maintained s 3° $m^{-1}$, and the patterns were recorded in the 2θ range of 10°–70° using Cu Kα radiation (λ = 1.5416 Å). The detailed structural analysis was further analyzed using FTIR and Raman spectroscopy. FTIR spectra were recorded using (Shimadzu Corporation, Kyoto, Japan) an IR-Prestige 21 spectrometer in the frequency range 400 to 4000 $cm^{-1}$ using KBr as a diluting agent. Raman spectra were examined by a commercial (Renishaw, Gloucestershire, UK) spectrometer equipped with a He-Ne laser (514.5 nm).

Morphological images were captured by an FESEM (Carl Zeiss Microscope Ltd., Germany) instrument equipped with an energy dispersive X-ray spectroscope. The Brunauer–Emmett–Teller (BET) test was performed to analyze the surface area, pore volume and pore size distribution using a commercial (Quantachrome Instruments, Boynton Beach, FL, USA) instrument. The prepared sample was degassed at 200 °C for 4 h prior to the nitrogen adsorption–desorption measurement. Thermogravimetric analysis (TGA) was performed to find the thermal stability of the samples using (Discovery STD-650, New Castle, DE, USA) TA Instruments.

The optical properties were recorded using a UV-vis spectrometer (Perkin Elmer, Waltham, MA, USA) in the range of 200 to 800 nm using a photoluminescence spectrofluorometer (Shimadzu, Kyoto, Japan) with an excitation wavelength of 360 nm. The electrochemical analysis was performed by (Novo Control, Montabaur, Germany) CH instrument. XPS analysis was performed using (PHI 5000, Chanhassen, MN, USA), and TEM images were obtained using a (FEI, Hillsboro, OR, USA) transmission electron microscope.

## 2.4. Photocatalytic Experiment

Degradation of RhB and MB

The photocatalytic test was examined by monitoring the degradation rate of dyes (RhB and MB). The test was performed in a photoreactor, and visible light irradiation was focused on it from the top. In order to perform the photocatalytic experiment, 100 mL of dye water solution was prepared from 10 ppm of dye solution in deionized water, taken in a quartz glass beaker (which allowed the penetration of light through it), and 20 mg of each photocatalyst was added to the dye solution. The dye–catalyst mixture was continuously stirred for an hour using a magnetic stirrer in dark conditions to allow the perfect adsorption of dye molecules on the photocatalyst's surface.

The rotation speed of the magnetic stirrer was set at 700 rpm, and the xenon lamp of 300 W employed with UV cut-off filter was maintained at a distance of 13 cm from the photoreactor. After every ten minutes, 3 to 5 mL aliquots were taken out and centrifuged and monitored with a UV-vis spectrometer. The degradation efficiency of the prepared photocatalysts was recorded after every ten minutes of the reaction. The percentage of degradation was calculated as (C/Co)*100, where C is the intensity of the concentration of the dye at each irradiated time interval and $C_0$ is the intensity of the initial concentration of the dye before irradiation. The photocatalytic degradation rate was calculated from the UV-vis absorbance plot. The rate of photoconversion of RhB to Rh-110 was recorded by observing the difference in intensity of absorbance peaks at 554 and 498 nm.

## 2.5. Electrochemical Analysis

Photoelectrochemistry was performed on the $Ag/Bi_2WO_6$ plasmonic structure, taken with a working electrode. An electrochemical workstation (CHI660) with athree-electrode configuration was used to record the electrochemical impedance spectra in the frequency range of $0.01–10^5$ Hz with the amplitude 5 mV.

In a typical procedure, 1 M NaOH in a 100 mL beaker was employed as an electrolyte, with a platinum wire as the counter electrode and Ag/AgCl as the reference electrode. A light source of a 300 W xenon lamp was used. The working electrode ($Ag/Bi_2WO_6$) was prepared by adding 8 mg of the prepared composite in 0.5 mL Dupont-Nafion to form a slurry. This prepared slurry was coated on a 2.5 mm × 2.5 mm × 1.1 mm fluorine-doped tin oxide glass (FTO) and stabilized in the refrigerator for 24 h.

## 3. Results and Discussions

### 3.1. Structural Study

Figure 3 shows the diffraction peak of pure $Bi_2WO_6$ and $Ag/Bi_2WO_6$ (1:1, 1:2 and 2:1) at room temperature. Considering the orthorhombic symmetry of the material, the characteristic diffraction peaks are indexed according to JCPDS No. 73-1126 [32] (Figure 3). No separate peaks for metallic silver were detected, which might be due to the low content of the metal. A sharp, well-defined diffraction peak corresponds to the crystalline nature of the catalysts.

The chemical composition of the prepared $Bi_2WO_6$ and $Ag/Bi_2WO_6$ (1:1, 1:2 and 2:1) was studied by FTIR spectra. The $Bi_2WO_6$ has an absorption band at $500–1000$ cm$^{-1}$. Figure 4 shows the stretching modes of Bi-O and W-O at $578.64$ cm$^{-1}$ and $732.95$ cm$^{-1}$, respectively [33,34]. The peaks at $3464.15$ cm$^{-1}$ and $1624.06$ cm$^{-1}$ are ascribed to O-H bending and stretching vibration of adsorbed $H_2O$ molecules, respectively [33,35]. No separate peaks for Ag were observed; this might be due to the lower content of Ag. Although, it is seen in the spectra that the intensity of the peak at $1381.03$ cm$^{-1}$ increases and the peak at $578.64$ cm$^{-1}$ decreases with Ag loading.

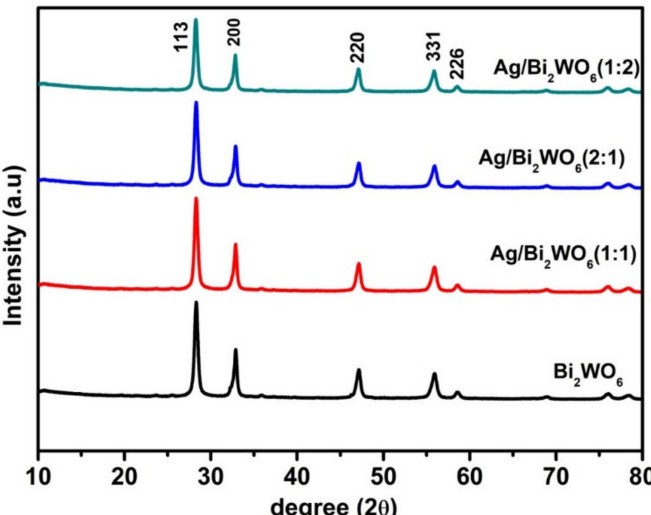

**Figure 3.** XRD spectra of pure $Bi_2WO_6$ and $Ag/Bi_2WO_6$ (1:1, 1:2 and 2:1).

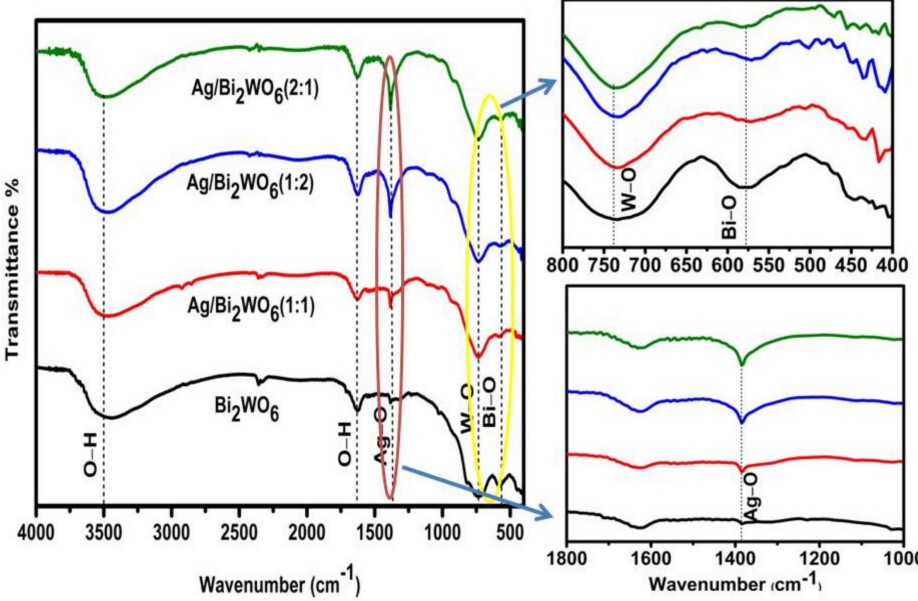

**Figure 4.** FTIR spectra of pure $Bi_2WO_6$ and $Ag/Bi_2WO_6$ (1:1, 1:2 and 2:1).

The decreased intensity peak at 578.64 cm$^{-1}$ is due to a decrease in the per unit volume of the functional group associated with the Bi-O bond. Photodeposition of Ag on $Bi_2WO_6$ prevents the IR radiation from reaching the molecule and hence reduces the absorption of light by Bi-O bond. The incorporation of Ag into the $Bi_2WO_6$ nanoparticles leads to the emergence of a new dip at 1381 cm$^{-1}$, representing the Ag-O bond [36]. After the silver coating, there was no change in the position of the $Bi_2WO_6$ peaks. The peaks have not changed, indicating that Ag has not destroyed the structure of $Bi_2WO_6$. Another interesting thing was noticed at peak 732.95 cm$^{-1}$ associated with the stretching mode of W-O; it became narrower with an increase in a molar ratio of Ag loading in $Bi_2WO_6$. This depicts that the photo-induced deposition of Ag on $Bi_2WO_6$ modified the response of $Bi_2WO_6$ in the IR region.

The surface morphologies of prepared $Ag/Bi_2WO_6$ were observed by FE-SEM images. The hydrothermal synthesis and its temperature played an important role in obtaining the crystalline and porous nano-flakes of the prepared composite. Figure 5a,b shows the FESEM image of $Ag/Bi_2WO_6$ (1:2). The numerous square nano-flakes of length ~200 nm are

observed in the micrograph. The circular colonies of various nano-flakes can be observed in Figure 5b. The nano-flakes self assembles themselves in the form of circular colonies.

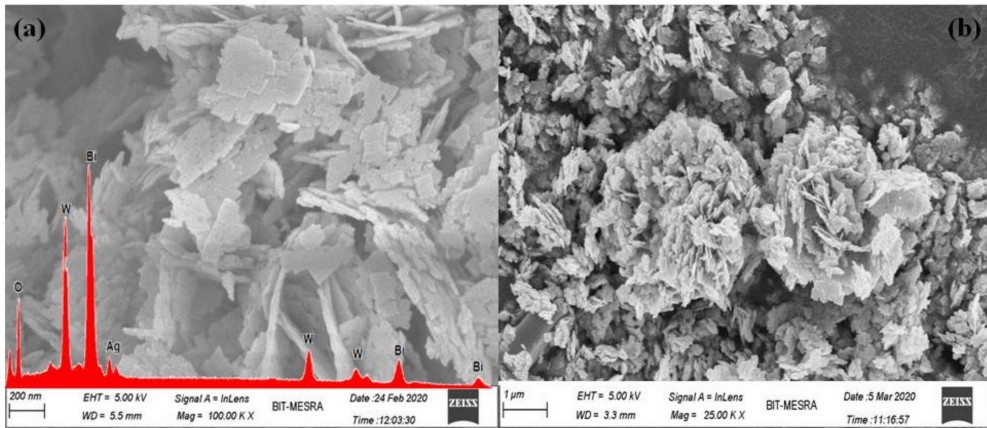

**Figure 5.** FESEM images of Ag/Bi$_2$WO$_6$ (1:2). (**a**) nanosheets are visible at 100.00 k × magnification; (**b**) circular colonies are visible at 25.00 K × magnification.

The Ag content is not visible in the images, which may be due to the very small sizes of silver nanoparticles. The pores and the crystallinity of the observed nano-flakes favor the adsorption of organic compounds and the transfer of active species [37,38]. There was no appreciable difference observed in FESEM images for Ag/Bi$_2$WO$_6$ (1:1), Ag/Bi$_2$WO$_6$ (1:2) and Ag/Bi$_2$WO$_6$ (2:1). The inset spectra in Figure 5a shows that the material is composed of Bi, W, O and Ag elements indicating the existence of Ag with Bi$_2$WO$_6$. The FESEM image and EDAX spectra of pure Bi$_2$WO$_6$ and Ag/Bi$_2$WO$_6$ (1:1 and 2:1) are not shown for briefness.

More detailed insights into the morphology of Ag/Bi$_2$WO$_6$ (1:1, 1:2 and 2:1) composites were investigated by TEM. The images of individual Bi$_2$WO$_6$ and Ag/Bi$_2$WO$_6$ composites with different molar contents of Ag to Bi$_2$WO$_6$ NPs are shown in Figure 6a–d. The selected area electron diffraction (SAED) pattern (Figure 6f) appears as bright concentric circles, which can be indexed to the (131), (200), (202), (331) and (262) planes of the Aurivillius-type layered structure Bi$_2$WO$_6$ [39].

The TEM images of Bi$_2$WO$_6$ and Ag/Bi$_2$WO$_6$ with different contents of Ag loaded are similar in size. It is clearly observed from Figure 6b–d that the greater density of the Ag NPs on the surface of nanosheets is found in the composite with a higher content of Ag (Ag/Bi$_2$WO$_6$ (2:1)). The darker region in the TEM images represents the area of Ag as it is the area of high electron density. The closer TEM image of the Ag/Bi$_2$WO$_6$ (2:1) (Figure 6e) clearly shows lattice spacing of 0.31 nm, which corresponds with (113) lattice planes of Bi$_2$WO$_6$, and the lattice fringes of 0.25 nm match well with the (111) plane of Ag. This result further proves the successful preparation of Ag/Bi$_2$WO$_6$ composite [40].

Figure 7 shows the TGA curve of Ag/Bi$_2$WO$_6$ (1:1, 1:2 and 2:1) in the temperature range of 25 °C to 800 °C. The maximum weight loss of 9% is observed in Ag/Bi$_2$WO$_6$ (2:1). The reason for weight loss is due to the decomposition of silver metal; silver nanoparticles decompose and lose weight between temperatures 200 °C to 450 °C [41]. The weight loss between 25 °C to 200 °C is due to the evaporation of moisture adsorbed from the atmosphere before performing the test. Ag/Bi$_2$WO$_6$ (1:1) underwent a weight loss of 7%, and Ag/Bi$_2$WO$_6$ (1:2) demonstrated a weight loss of 4% within the given temperature range.

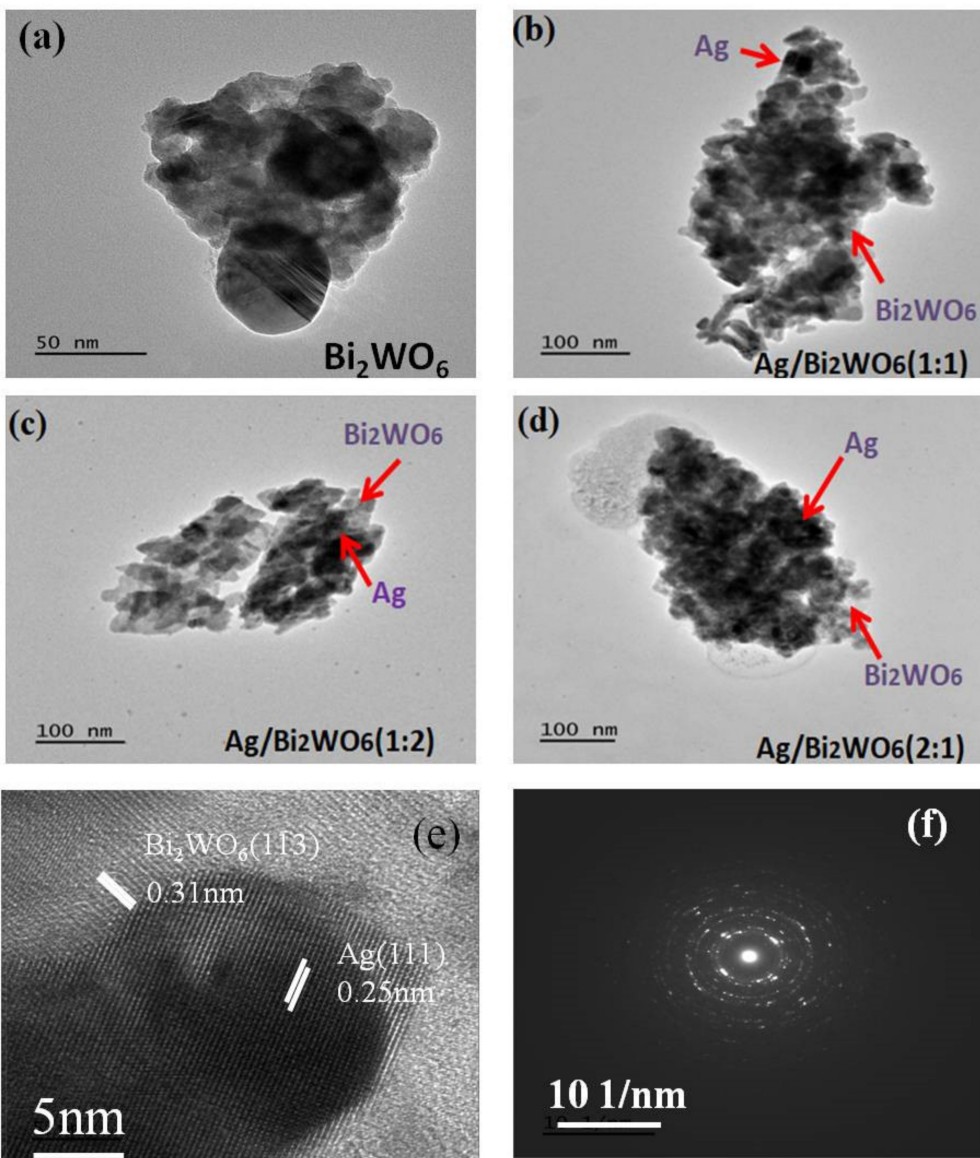

**Figure 6.** TEM images of (**a**) $Bi_2WO_6$, (**b**) $Ag/Bi_2WO_6$ (1:1), (**c**) $Ag/Bi_2WO_6$ (1:2) and (**d**) $Ag/Bi_2WO_6$ (2:1); (**e**) lattice spacing of $Ag/Bi_2WO_6$ (1:2); and (**f**) SAED pattern of $Ag/Bi_2WO_6$ (1:2).

X-ray photoelectron spectroscopy (XPS) was performed on pure $Bi_2WO_6$ and $Ag/Bi_2WO_6$ (1:1, 1:2 and 2:1) as shown in Figure 8. The binding energy in the spectrum was calibrated using that of C1s (284.62 eV). Figure 8 is the overall XPS spectrum of the $Bi_2WO_6$ and $Ag/Bi_2WO_6$ (1:1, 1:2 and 2:1) heterostructure. No peak corresponding to Ag is detected in the overall XPS spectrum of $Bi_2WO_6$, whereas composites of $Ag/Bi_2WO_6$ show Ag peaks, indicating that the photodeposition method is successful for Ag deposition. The peaks centering in the region 159.23 and 164.57 eV (Figure 9a) can be designated to be the binding energies of Bi $4f_{7/2}$ and Bi $4f_{5/2}$ in $Bi^{3+}$ [42].

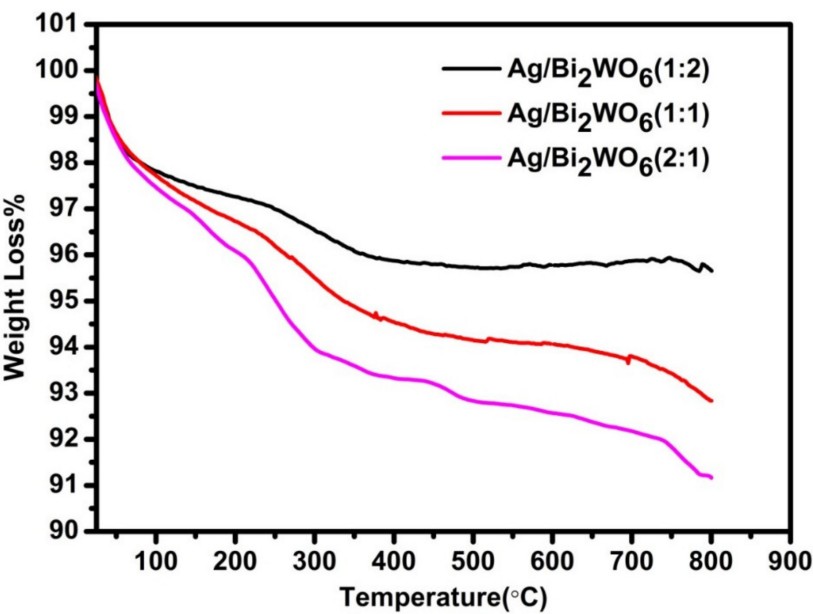

**Figure 7.** TGA spectra of Ag/Bi$_2$WO$_6$ (1:1, 1:2 and 2:1).

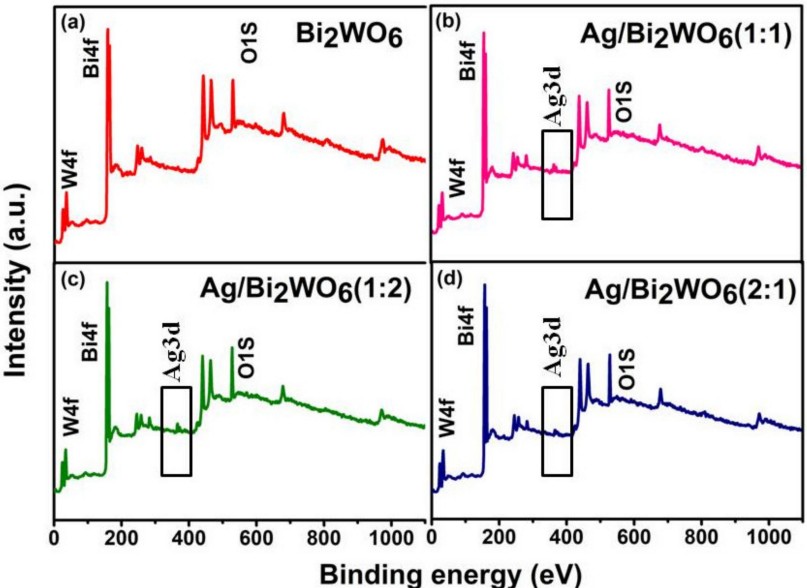

**Figure 8.** Overall XPS spectra of (**a**) Bi$_2$WO$_6$; (**b**) Ag/Bi$_2$WO$_6$ (1:1); (**c**) Ag/Bi$_2$WO$_6$ (1:2); (**d**) Ag/Bi$_2$WO$_6$ (2:1).

The peaks centering in the region 35.44–37.59 eV (Figure 9b) can be ascribed to W 4f$_{5/2}$ and W 4f$_{7/2}$ in the W$^{6+}$ oxidation state. Ag/Bi$_2$WO$_6$ (1:2) shows the largest positive binding energy shift of 0.3 and 0.4 eV, depicting a higher oxidation state of W in the case of Ag/Bi$_2$WO$_6$ (1:2). This is due to the higher interaction of Bi$_2$WO$_6$ with Ag [42,43]. All the measured values are consistent with the previous reports [44,45]. The peaks centering at 373.74 and 367.72 eV (Figure 9c) can be ascribed to Ag 3d$_{3/2}$ and Ag 3d$_{5/2}$ [46].

Considering the binding energy of Ag 3d$_{3/2}$ and Ag 3d$_{5/2}$, the valence of Ag in the heterostructure can be identified to be +1. The positive shift of Ag in Ag/Bi$_2$WO$_6$ (1:2) composite is evidence of the strong interaction between Ag and Bi$_2$WO$_6$ [47]. The binding energy of O1S (Figure 9d) lies at 530.20 eV and there is a large negative shift observed in the case of the binding energy of O1S in Ag/Bi$_2$WO$_6$ (1:2) composite. The strong interaction of the composite with Ag creates an electric field; this weakens the bond and O atoms in Bi$_2$WO$_6$ are replaced creating oxygen vacancies [42,43]. The increased surface oxygen

vacancy decreased the surface recombination centers and improved the charge separation efficiency, thus, enhancing the photocatalytic activity [48].

Figure 10a,b shows the N$_2$ adsorption–desorption isotherm and the corresponding pore size distribution of the prepared photocatalysts. The BJH (Barett, Joyner and Halenda method) pore size distribution plot of prepared Bi$_2$WO$_6$ and Ag/Bi$_2$WO$_6$ (1:1, 1:2 and 2:1) shows a narrow range of pore size distribution with average pore diameters of 15.18, 8.70, 8.69 and 8.73 nm, respectively, indicating the mesoporous characteristic of the photocatalyst. Table 2 gives the specific surface area, pore diameter and pore volume of the Bi$_2$WO$_6$ and Ag/Bi$_2$WO$_6$ with a different molar ratio of Ag to Bi$_2$WO$_6$. There is no appreciable difference in surface area noticed.

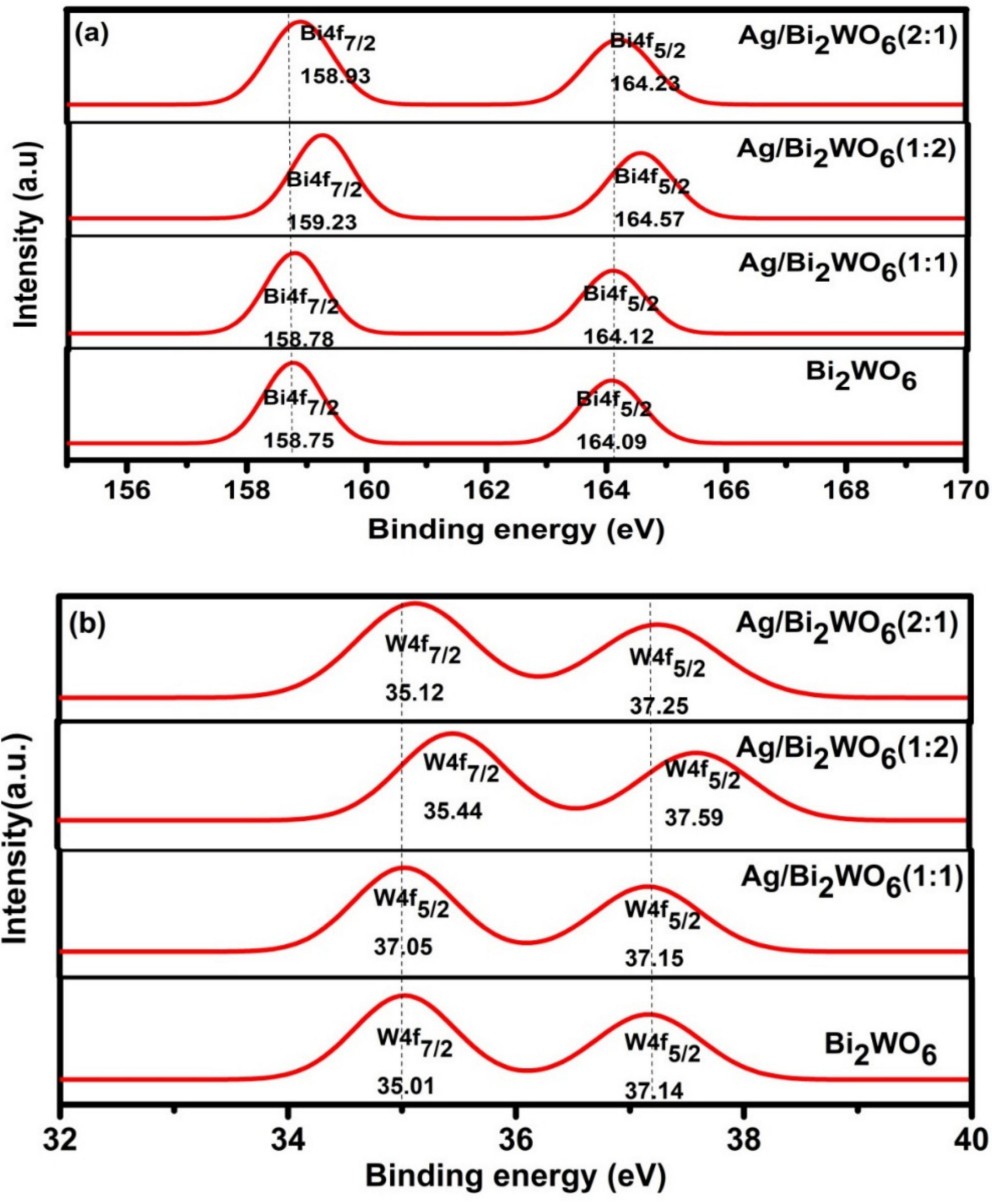

**Figure 9.** *Cont.*

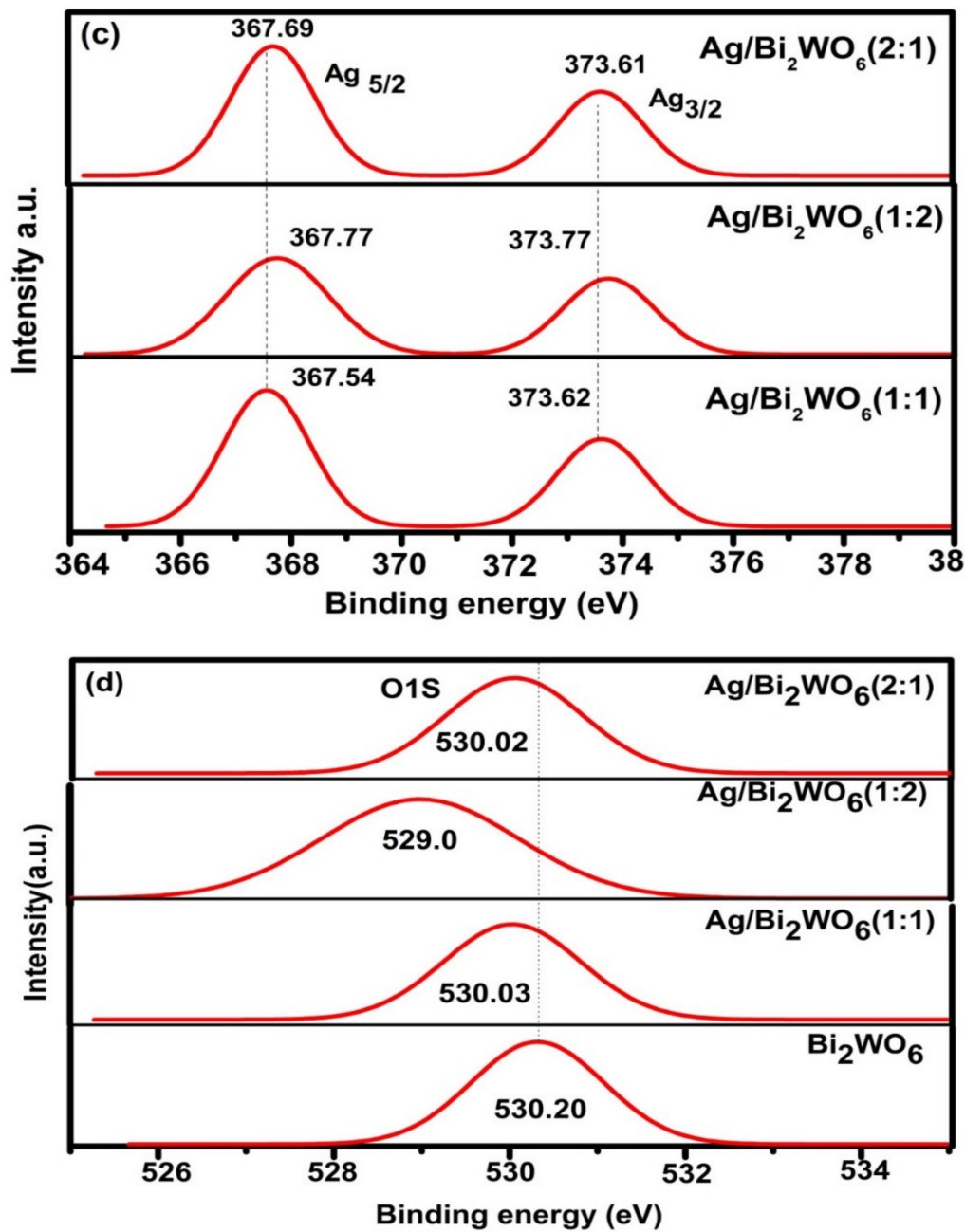

**Figure 9.** XPS spectra: (**a**) Bi4f peaks, (**b**) W4f peaks, (**c**) Ag3d peaks and (**d**) O1S peaks.

**Table 2.** Summary of BET result of pure $Bi_2WO_6$ and $Ag/Bi_2WO_6$ (1:1, 1:2 and 2:1).

| Catalyst | Specific Surface Area (m²/g) | Pore Diameter (nm) | Pore Volume (cc/g) |
|---|---|---|---|
| $Bi_2WO_6$ | 26.40 | 15.18 | 0.12 |
| $Ag/Bi_2WO_6$ (1:1) | 26.02 | 8.70 | 0.11 |
| $Ag/Bi_2WO_6$ (1:2) | 20.84 | 8.69 | 0.07 |
| $Ag/Bi_2WO_6$ (1:1) | 20.43 | 8.73 | 0.12 |

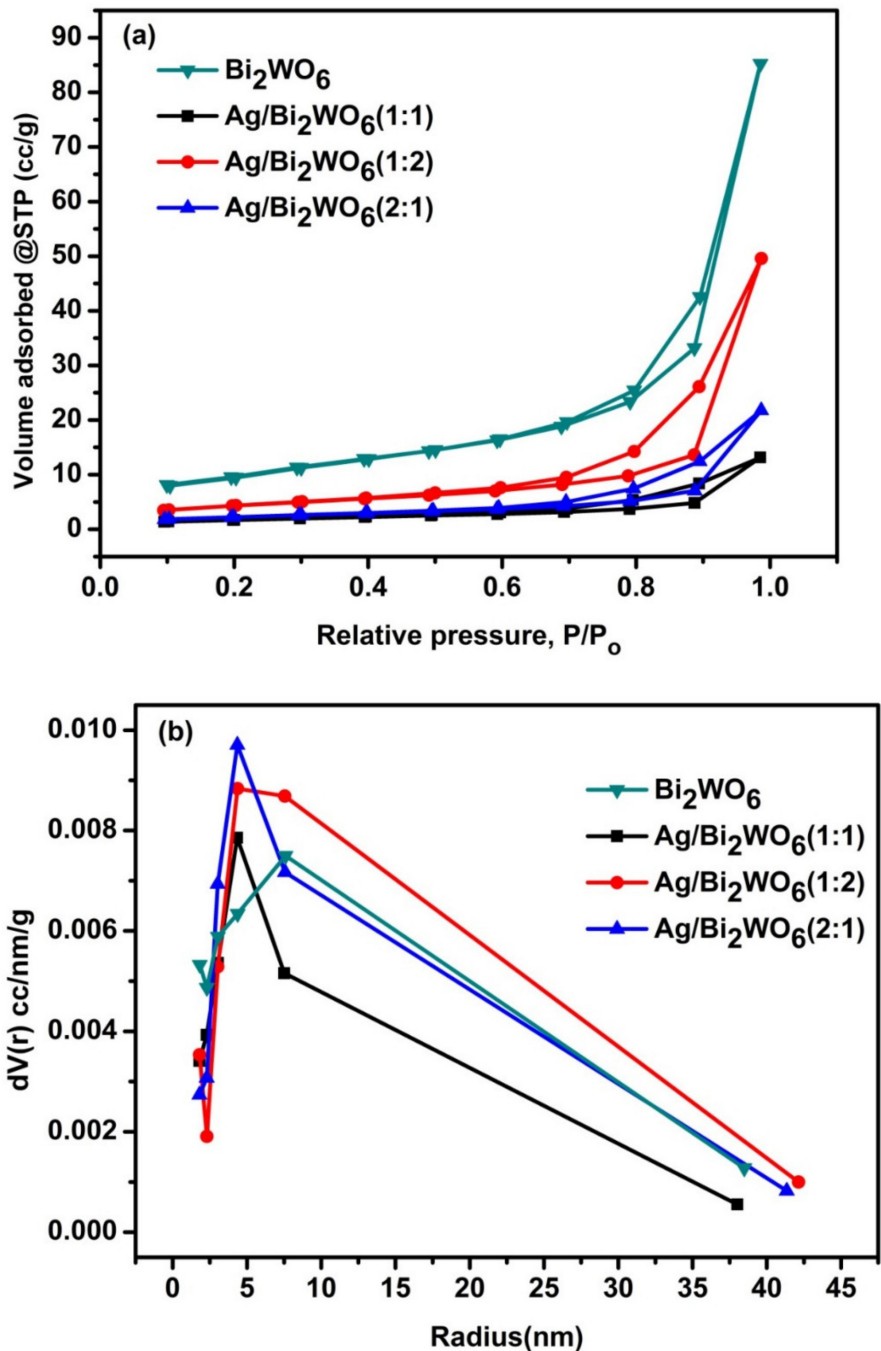

**Figure 10.** (**a**) $N_2$ adsorption–desorption isotherm. (**b**) Pore size distribution of pure $Bi_2WO_6$ and $Ag/Bi_2WO_6$ (1:1, 1:2 and 2:1).

### 3.2. Electrochemical Properties

The charge transfer process in $Ag/Bi_2WO_6$ composite with different molar ratio are demonstrated by the electrochemical impedance spectroscopy (EIS) measurements. Figure 11 shows the Nyquist plots of $Ag/Bi_2WO_6$ (1:1, 1:2 and 2:1) in both dark and light conditions. The smaller diameter of the semicircle of the Nyquist plot suggests the effective separation of photogenerated electron–hole pair in the materials. In the present EIS measurement, we found that the arc radius of $Ag/Bi_2WO_6$ (1:2) was much smaller than that of the $Ag/Bi_2WO_6$ (1:1 and 2:1). This indicates that the LSPR effect of $Ag/Bi_2WO_6$ strengthens the charge transportation and weakens the recombination rate, which is in good agreement with the photoluminescence and UV absorbance result.

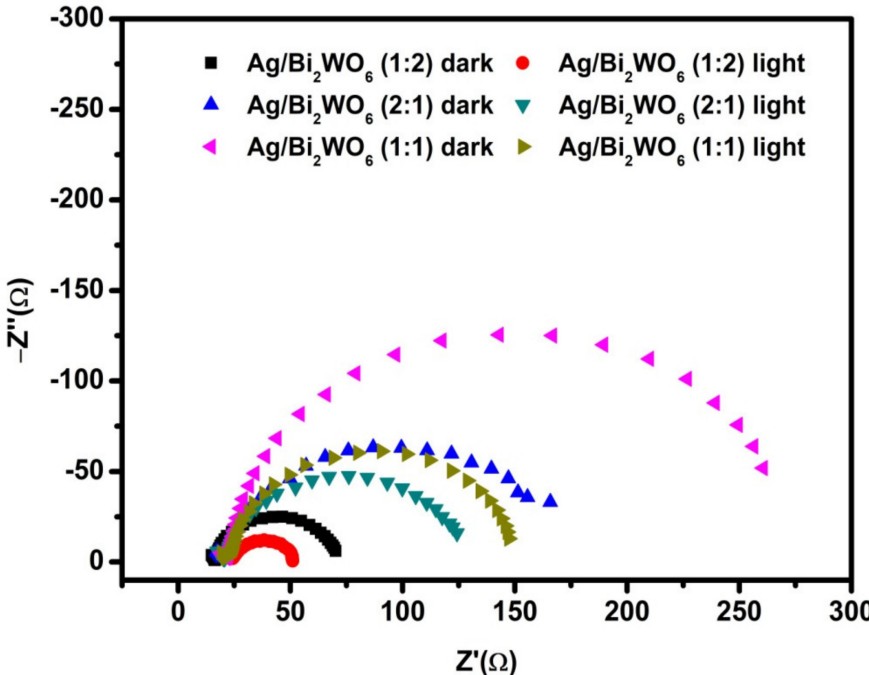

**Figure 11.** Nyquist plot of Ag/Bi$_2$WO$_6$ (1:1), Ag/Bi$_2$WO$_6$ (1:2) and Ag/Bi$_2$WO$_6$ (2:1) in dark and light conditions.

*3.3. Optical Properties*

The optical properties of the prepared plasmonic photocatalysts are observed through a UV-vis spectrometer. The absorbance spectra of Bi$_2$WO$_6$, Ag/Bi$_2$WO$_6$ (1:1, 1:2 and 2:1) are depicted in Figure 12a. According to the absorption spectra, as shown, the pure Bi$_2$WO$_6$ sample exhibits strong photoresponse properties from the UV light region to visible light shorter than 430 nm due to the intrinsic bandgap transition. The synergistic effect of the LSPR with the light absorption promotes the absorption of Ag/Bi$_2$WO$_6$ (1:2) composites extends over a wider Vis-light region as compared to Ag/Bi$_2$WO$_6$ (1:1) and Ag/Bi$_2$WO$_6$ (2:1) [49,50].

The Ag-loaded Bi$_2$WO$_6$ (1:2) composites show an enhanced photo-absorption property in the visible light region. Ag/Bi$_2$WO$_6$ (1:1) shows that a decrease in absorption peak may be due to large crystal vacancies and larger carrier density. The enhancement in the absorption peak may be attributed to the SPR effect [51–53]. The tau plot calculated the optical band gap of pure Bi$_2$WO$_6$ and Ag/Bi$_2$WO$_6$ (1:1, 1:2 and 2:1), which is found to be 3.06, 2.71, 2.41 and 2.85 eV, respectively (Figure 12b). The optical band gap of the Ag/Bi$_2$WO$_6$ (1:2) nano photocatalyst is calculated to be 2.41 eV, which is less compared to Bi$_2$WO$_6$ and Ag/Bi$_2$WO$_6$ (1:1 and 2:1). This result is in accordance with the XPS result, which showed that higher oxygen vacancy had been created in Ag/Bi$_2$WO$_6$ (1:2) catalyst. The surface plasmon resonance on the surface of the silver is due to the oscillation of free electrons excited by the light of a matching frequency.

The electron trapping procedure by Ag NPs can be explained by photoluminescence (PL) emission spectra of Ag/Bi$_2$WO$_6$. The PL emission spectra in Figure 12c show the decrease in PL intensity as the Ag loading increases. Since recombination of electrons and hole pairs is a radiative process that is ascribed to PL emission, the intensity of the PL spectra reduces when the recombination process is suppressed [39,40]. This suggests that the introduction of Ag NPs in Bi$_2$WO$_6$ can moderately restrain the combination of photogenerated holes and electrons.

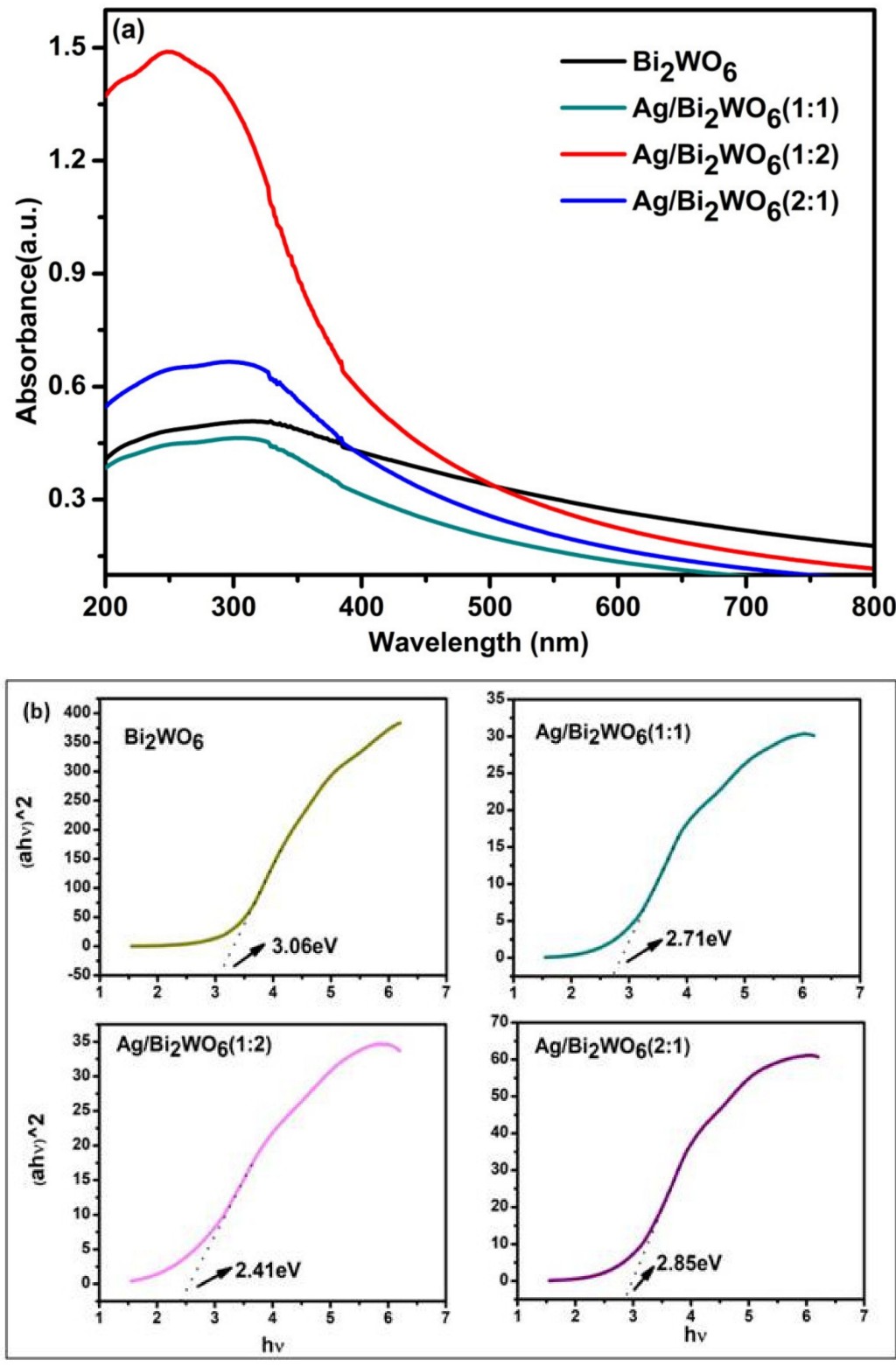

**Figure 12.** *Cont.*

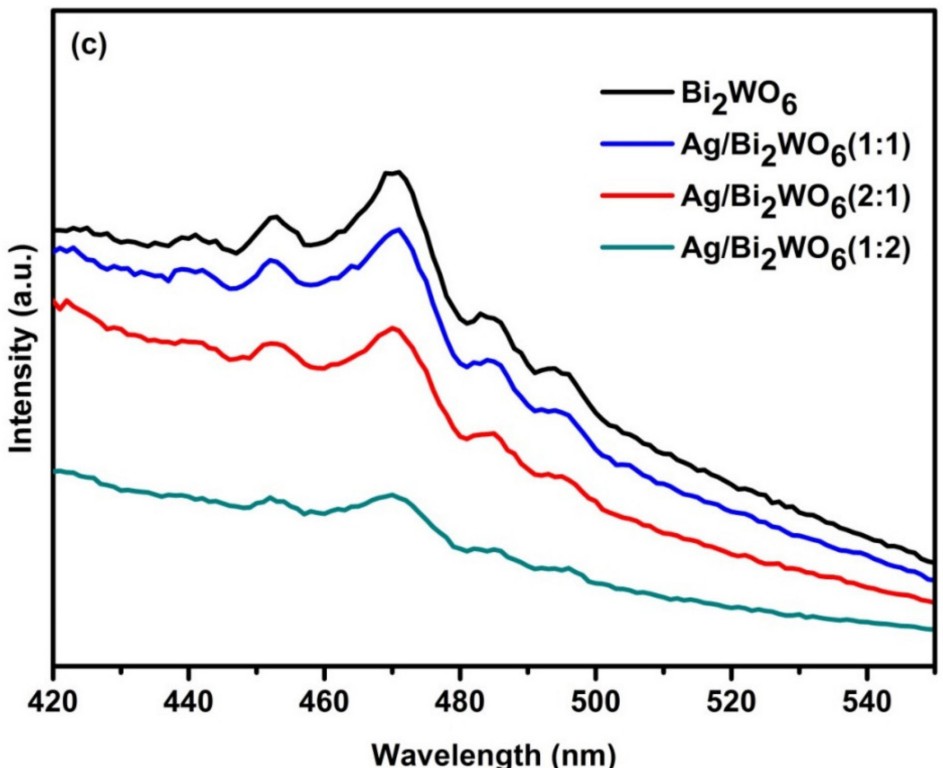

**Figure 12.** (**a**) Absorbance spectra, (**b**) optical band gap and (**c**) PL spectra of pure $Bi_2WO_6$ and $Ag/Bi_2WO_6$ (1:1, 1:2 and 2:1).

### 3.4. Photocatalytic Activity of RhB and MB

The photocatalytic activity of pure $Bi_2WO_6$ and $Ag/Bi_2WO_6$ (1:1, 1:2 and 2:1) is performed over RhB and MB under visible light irradiation. The reaction was performed in a photoreactor under a 300 W xenon lamp with UV cut-off filter. The aliquots of the dye are monitored after every ten minutes by UV spectrometer, and we observed that the absorption peak of RhB at about 554 nm and MB at about 650 nm decreases gradually and shifts towards a smaller wavelength with time under visible light irradiation. The absorbance peak of RhB decreases and is followed by the absorption band shift from 554 nm to 498 nm.

The blue shift in the absorbance wavelength of RhB is due to the selective photocatalytic oxidation, which converts RhB to Rh-110 [54,55]. Pure $Bi_2WO_6$ and $Ag/Bi_2WO_6$ (1:1 and 2:1) converts 39%, 51% and 74% of RhB to Rh110 in 90 min (Figure 13a,b,d, respectively. As far as Vis-light irradiation is concerned, the maximum absorption wavelength (lmax) of RhB solution in the presence of $Ag/Bi_2WO_6$ (1:2) blue shifts from 554 to 498 nm in 90 min and rises almost equal to the initial peak intensity corresponding to the absorption peak intensity at 0 min (Figure 13c). This indicates the selective photocatalytic oxidation toward RhB [55].

At this phase, the conversion ratio to Rh-110 is ~83% and at the end of the photocatalytic oxidation reaction, the color of RhB changes to pale green, thus, confirming the formation of Rh-110 (Figure 13g). The breakdown pathway of RhB includes RhB at 554 nm forms N,N,N′-Triethyl-rhodamine at 539 nm, N,N′-Diethyl-rhodamine at 522 nm; N-Ethyl-rhodamine at 510 nm; and ultimately to Rh-110 at wavelength 498 nm [56]. The de-ethylation pathway in the breakdown of RhB to Rh-110 is shown in Figure 13f. $Ag/Bi_2WO_6$ (1:1) shows maximum RhB degradation of 70% in 40 min; however, the conversion ratio of RhB to Rh-110 is only 50%, whereas $Ag/Bi_2WO_6$ (1:2) shows 50% RhB degradation in 10 min and has a maximum RhB to Rh-110 conversion of about ~83% (Figure 13e).

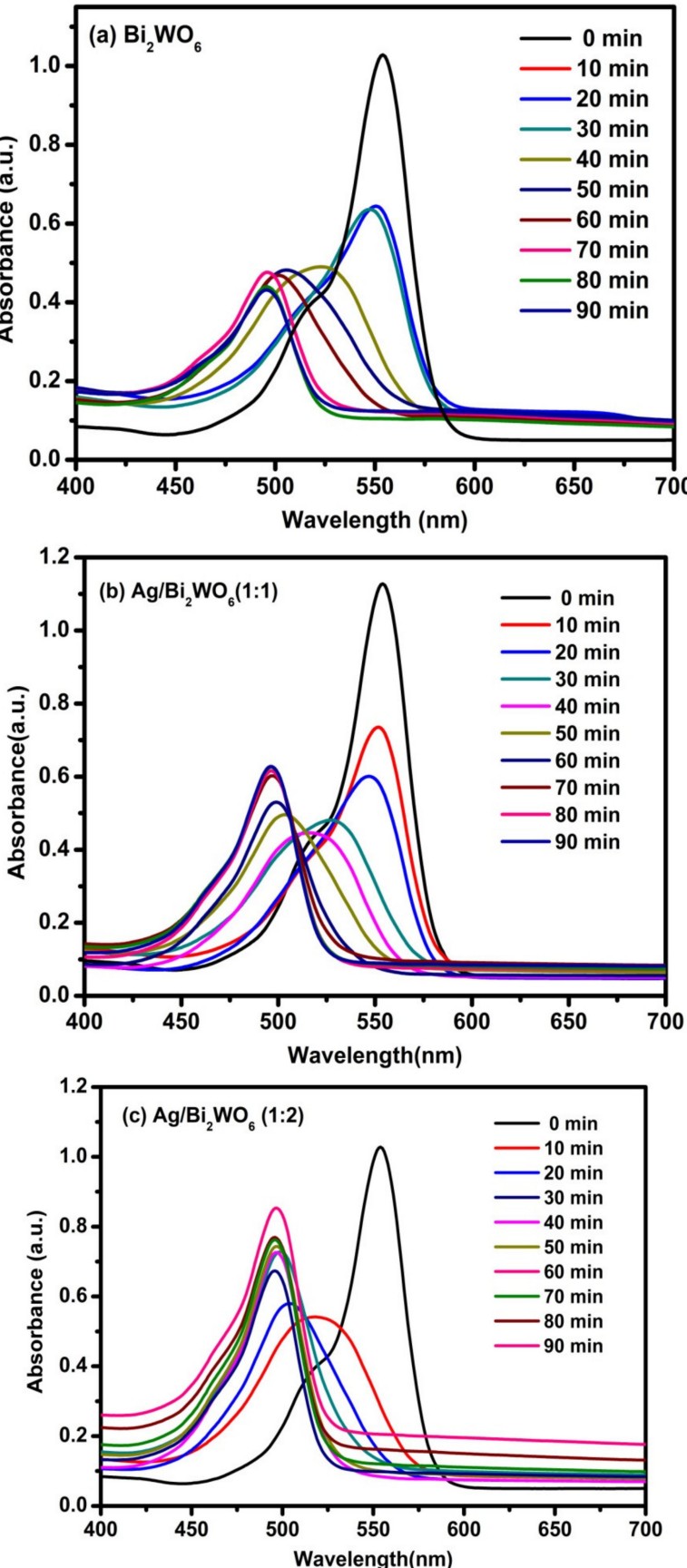

**Figure 13.** *Cont.*

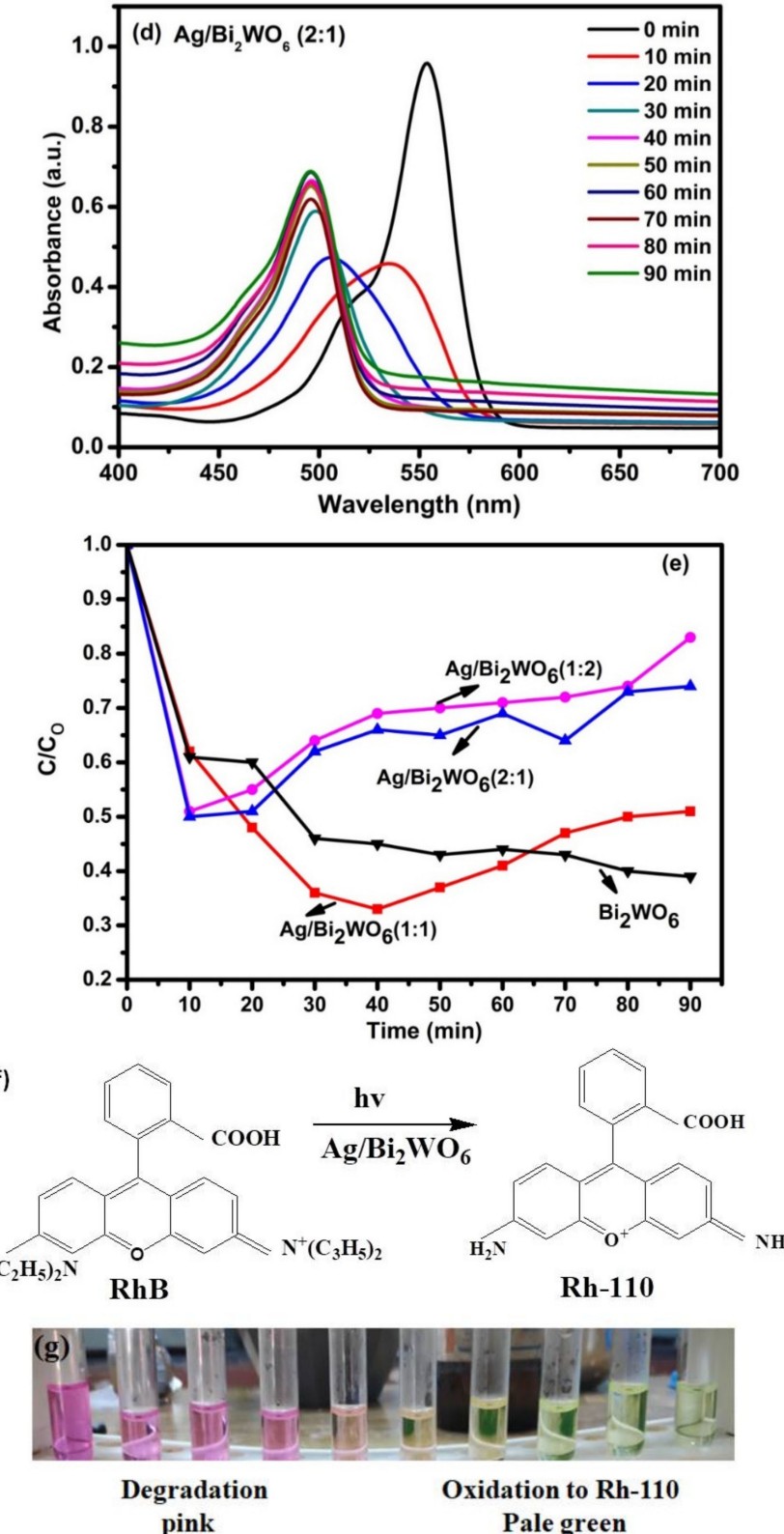

**Figure 13.** Photocatalytic oxidation of RhB by (**a**) Bi$_2$WO$_6$, (**b**) Ag/Bi$_2$WO$_6$ (1:1), (**c**) Ag/Bi$_2$WO$_6$ (1:2) and (**d**) Ag/Bi$_2$WO$_6$ (2:1); (**e**) photocatalytic degradation efficiency of the prepared photocatalysts; (**f**) conversion of RhB to Rh-110; and (**g**) images of RhB taken after every 10 min of the reaction.

A higher MB removal efficiency of 68% in 90 min was observed with Ag/Bi$_2$WO$_6$ (1:2) (Figure 14c). Pure Bi$_2$WO$_6$ and Ag/Bi$_2$WO$_6$ (1:1 and 2:1) degraded only 38%, 43%

and 53% of MB in 90 min (Figure 14a,b,d respectively; whereas bare $Bi_2WO_6$ could able to degrade only 38% of MB. This improved performance of $Ag/Bi_2WO_6$ (1:2) is attributed to the plasmonic effect created by the presence of the optimum amount of Ag content in the photocatalyst.

When a photocatalyst is illuminated by light, it might create a Schottky barrier at the junction of Ag and $Bi_2WO_6$ and generate a high-intensity electric field, which creates a space–charge region and allowed the separation of charges or the transfer of electrons from $Bi_2WO_6$ to the Ag. The creation of the Schottky barrier led to the charge transfer from $Bi_2WO_6$ to Ag NPs; here, Ag must have acted as an electron trapper. This reduced the recombination rate of electrons and holes. Thus, enhanced degradation efficiency was achieved by the photocatalyst $Ag/Bi_2WO_6$ (1:2). Photocatalysts of $Ag/Bi_2WO_6$ (1:1 and 2:1) could not produce such results due to the lack of electric field generation.

The XPS result has already shown that the maximum oxygen vacancy is created in the case of $Ag/Bi_2WO_6$ (1:2) and has the lowest optical band gap (~2.41 eV) among the other prepared catalyst, which indicates that the intense electric field is created in $Ag/Bi_2WO_6$ (1:2). The high intense electric field observed in the absorbance plot must have enhanced the charge separation efficiency of the catalyst. The PL result also suggests the lowest recombination rate of electron–hole pair; all these results support the higher photocatalytic effect of $Ag/Bi_2WO_6$ (1:2). There has not been much change in the surface area and the pore volume of the prepared photocatalysts. Thus, depicting that the surface area and porosity of the material is not the key factor here for the photocatalytic activity of the prepared plasmonic photocatalysts.

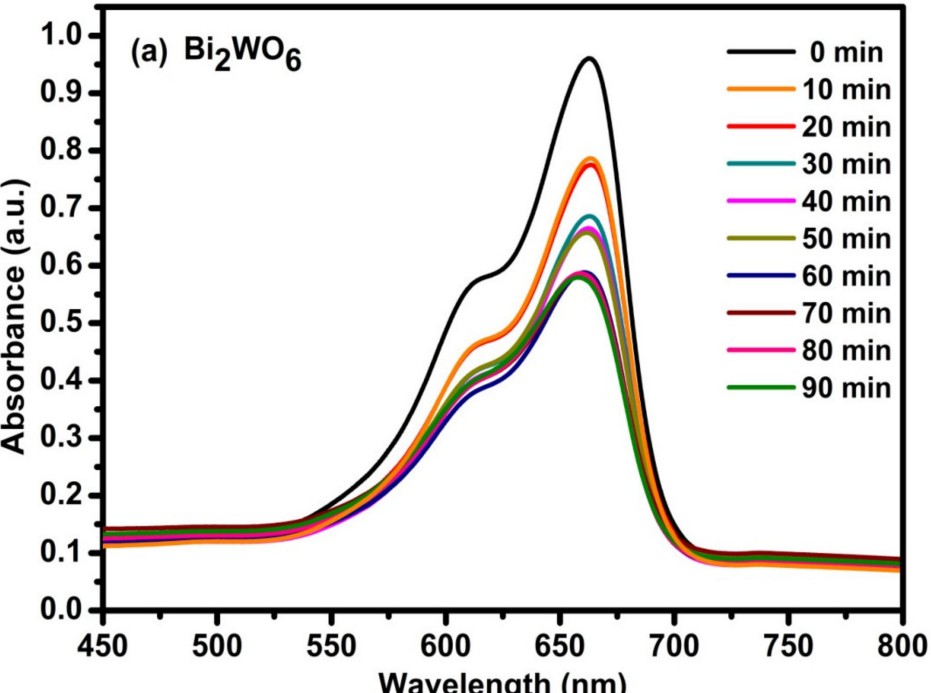

**Figure 14.** *Cont.*

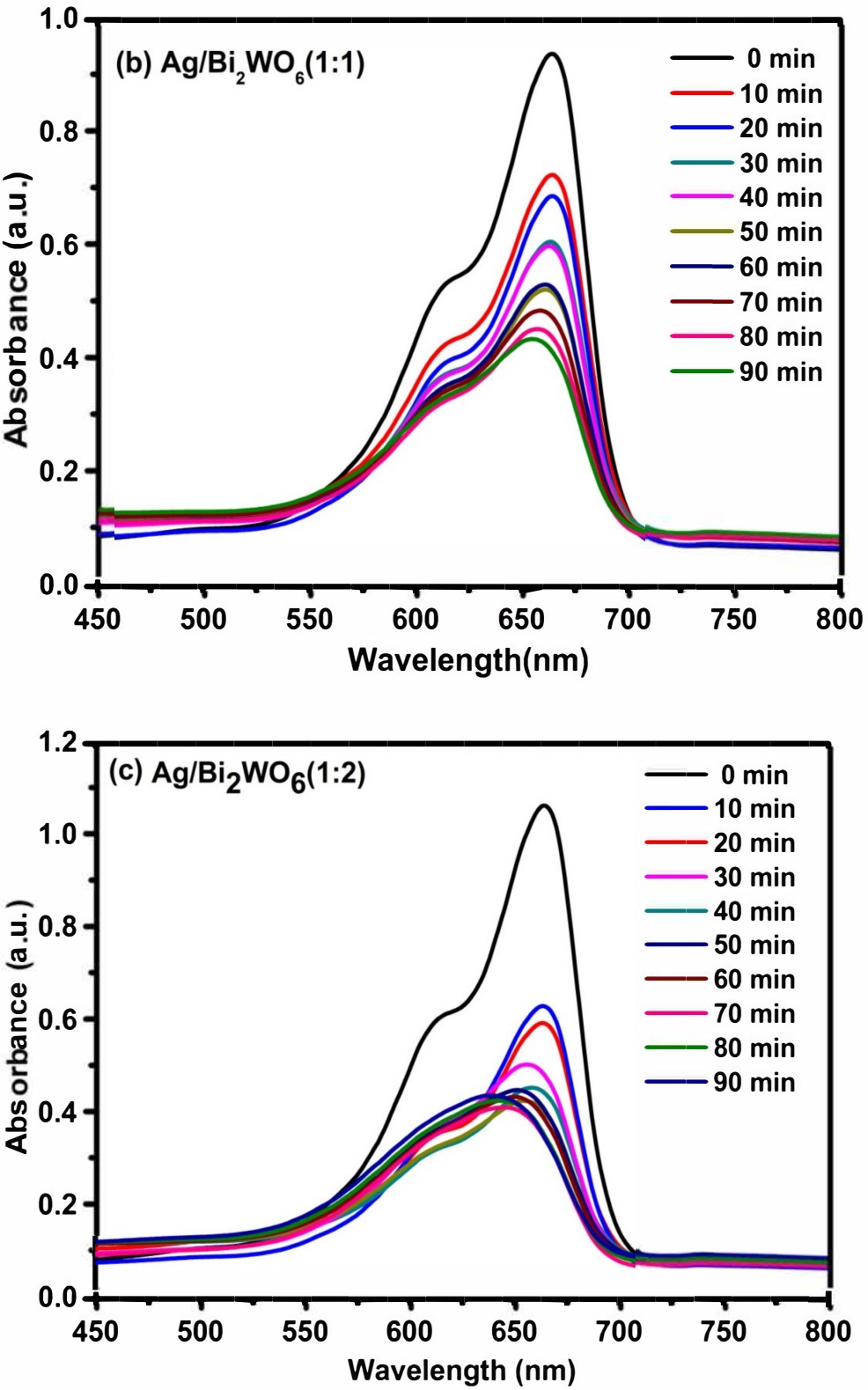

**Figure 14.** *Cont.*

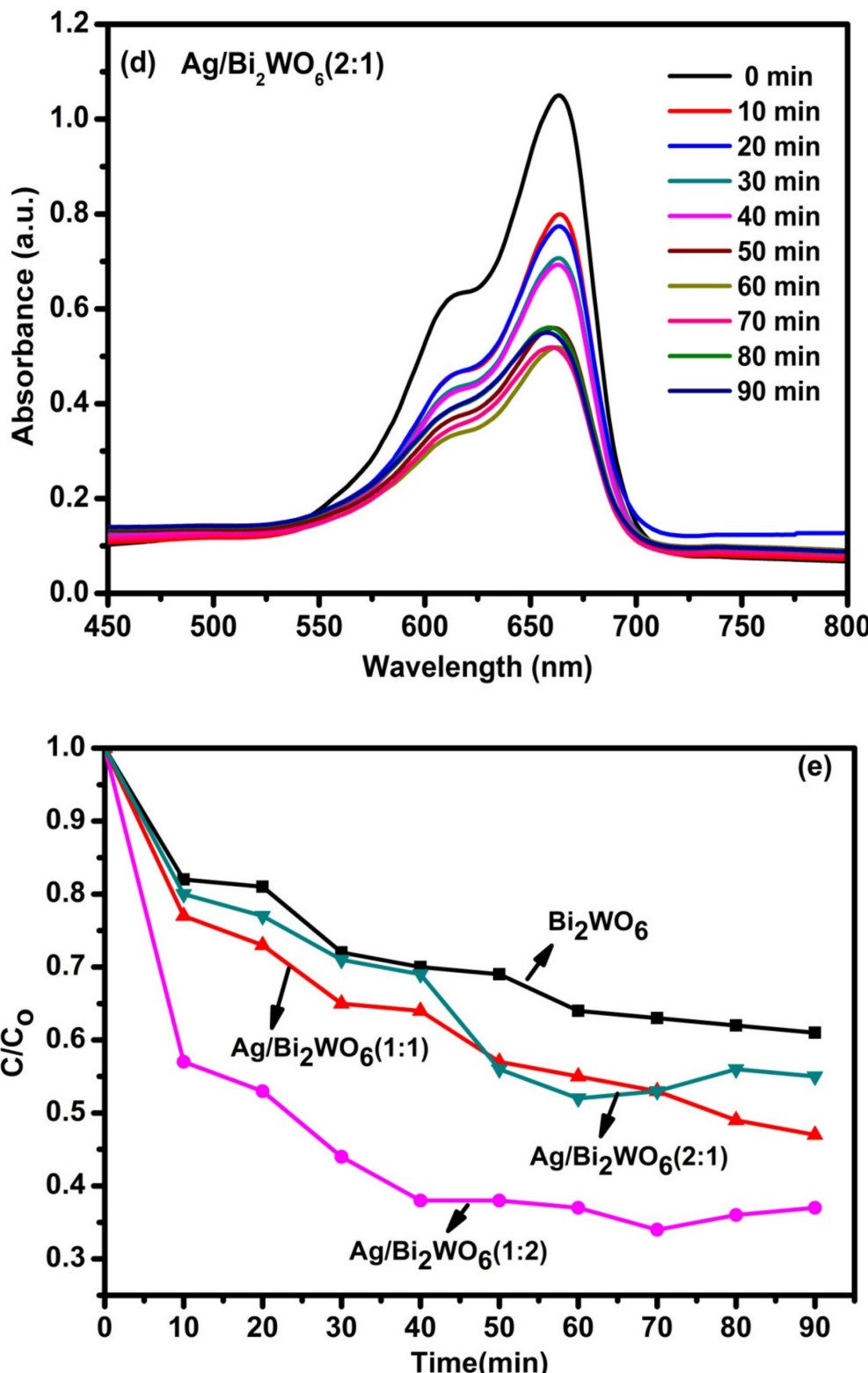

**Figure 14.** Photocatalytic degradation of MB by (**a**) $Bi_2WO_6$, (**b**) $Ag/Bi_2WO_6$ (1:1), (**c**) $Ag/Bi_2WO_6$ (1:2), (**d**) $Ag/Bi_2WO_6$ (2:1) and (**e**) the photocatalytic degradation efficiency of the prepared composites.

*3.5. Proposed Mechanism*

In a typical photocatalytic reaction, the photocatalyst when illuminated, produces electron ($e^−$) and holes ($h^+$) pair, which is ultimately converted to various reactive oxygen species, such as hydroxyl radicals ($^*OH$), superoxide anion radicals ($^*O_2^−$), hydroperoxyl

radical (*OOH) and so on [57]. The mechanism involved in explaining the above results could be presented through the following reactions [54].

$$Ag/Bi_2WO_6 + hv \rightarrow (e^-(CB) + h^+(VB)) \tag{1}$$

$$H_2O + h^+ (VB) \rightarrow *OH + H^+ \tag{2}$$

$$OH^- + h^+ (VB) \rightarrow *OH \tag{3}$$

$$O_2 + e^- (CB) \rightarrow *O_2^- \text{ (superoxide radical)} \tag{4}$$

$$O_2^- + H^+ \rightarrow *OOH \text{ (hydroperoxyl radical)} \tag{5}$$

$$*OOH + *O_2^- + H^+ \rightarrow H_2O_2 + O_2 \tag{6}$$

$$H_2O_2 + *O_2^- \leftrightarrow *OH + OH^- + O_2 \tag{7}$$

$$Dye + *OH \rightarrow CO_2 + H_2O \text{ (dye intermediate products)} \tag{8}$$

$$Dye + h^+ (VB) \rightarrow \text{oxidation product} \tag{9}$$

$$Dye + e^- (CB) \rightarrow \text{Reduction product} \tag{10}$$

The schematic diagram of photocatalytic oxidation process of $Ag/Bi_2WO_6$ (1:2) under visible light irradiation is illustrated in Figure 15 Under visible light irradiation, a large amount of electrons hole-pair is generated due to the presence of $Bi_2WO_6$ and the LSPR effect of Ag. The electrons start moving from $Bi_2WO_6$ to Ag. Silver nanoparticles, in this case, act as an electron trapper and these energetic electrons are transferred to Ag NPs.

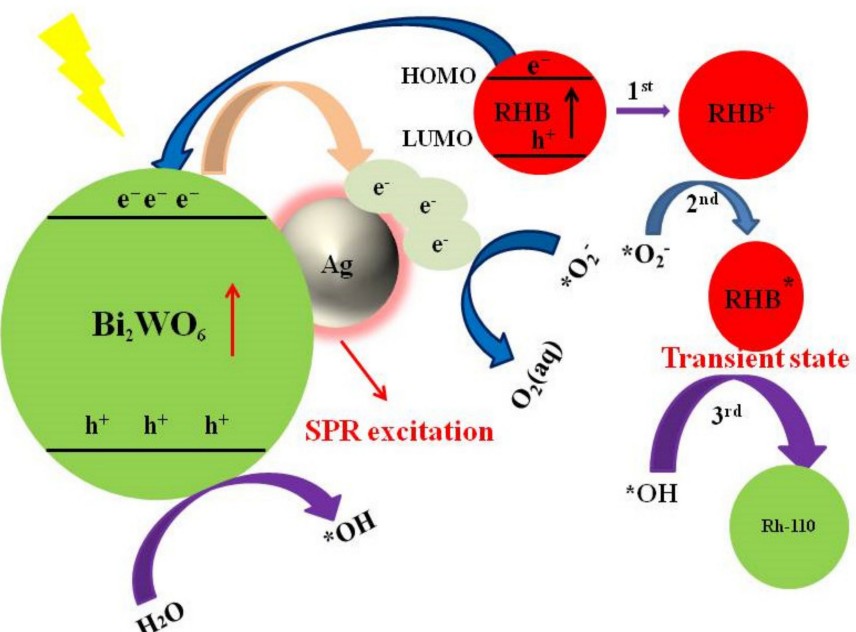

**Figure 15.** Schematic diagram of the possible process for $Ag/Bi_2WO_6$ composites photocatalytic degrading of RhB under visible light irradiation.

RhB is photosensitized and loses $e^-$ from a higher occupied molecular orbital and converts to RhB$^+$ [58]. According to Equation (4), *O$_2^-$ formed in the reaction reacts with the transient state of RhB*. The conjugated xanthene structure of RhB could not be destroyed by *O$_2^-$, because the potential of *O$_2^-$ is weaker than *OH. RhB$^+$ is oxidized by *OH to form Rh-110. In contrast, when the effect of *OH is greater than *O$_2^-$, *OH directly damages the xanthene structure of RhB, the transient structure of RhB is not produced and RhB-110 is not obtained.

Herein, Ag/Bi$_2$WO$_6$ (1:2) exhibited higher selective oxidation activity under a visible light source. Based on the equations given below, we propose that Ag/Bi$_2$WO$_6$ (1:2) can produce *O$_2$ and suppress the increase of *OH. In order to investigate the mechanism of photocatalytic activity of RhB, benzoquinone (BQ), triethanolamine (TEOA) and iso-propanol (IPA) were used to scavenge *O$_2$, h$^+$ and *OH and under visible light irradiation. According to Equations (2), (3) and (5), the effect of h$^+$ is ultimately converted to *O$_2$ and *OH.

Therefore, we considered only the effect of *O$_2$ and *OH in the scavenger test. The photocatalytic activity of Ag/Bi$_2$WO$_6$ (1:2) in the presence of BQ and IPA is shown in Figure 16. The blue shift in the absorbance wavelength is not observed and the degradation rate decreases when *O$_2$ is quenched. When IPA is used to quench *OH blue shift in the absorbance wavelength is observed, depicting that *O$_2$ plays an important role in the conversion of RhB to Rh-110.

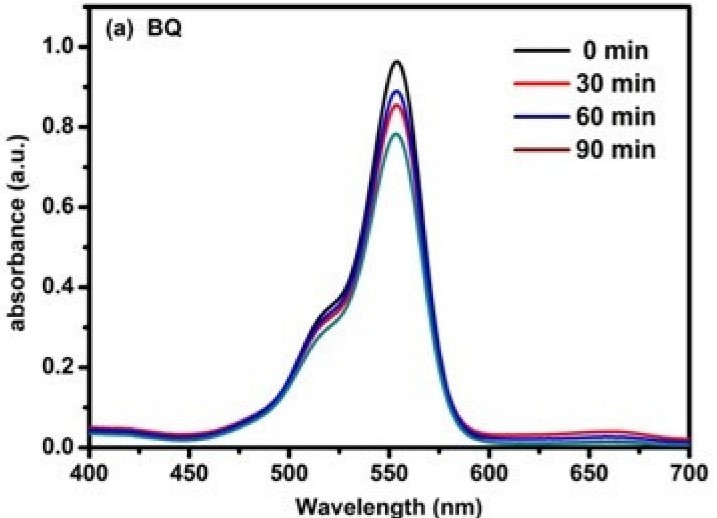

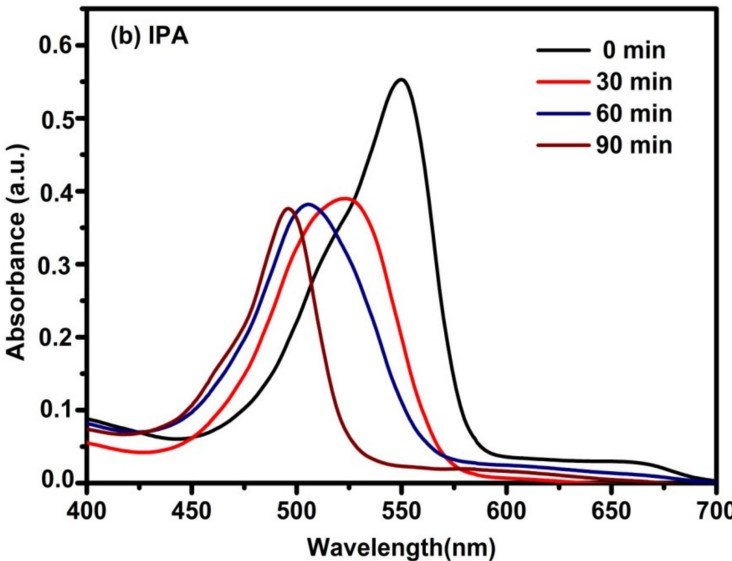

**Figure 16.** Absorbance spectra of RhB using photocatalyst Ag/Bi$_2$WO$_6$ (1:2) in the presence of scavengers. (**a**) BQ. (**b**) IPA under visible light irradiation.

## 4. Conclusions

In this paper, we designed plasmonic nanostructures of pure $Bi_2WO_6$ and $Ag/Bi_2WO_6$ (1:1, 1:2 and 2:1) photocatalysts with highly photocatalytic oxidation activity. The LSPR effect of Ag NPs on $Bi_2WO_6$ with a molar ratio of (1:2) composite improves the conversion ratio of RhB to Rh-110 to 83% but also degrades 68% of MB in 90 min under visible light irradiation. The combination of the noble metal with the photocatalyst introduced the plasmonic effect, which highly enhanced the photocatalytic activity of the photocatalyst. The scavenging experiment confirms the dominant effect of $*O_2$ in the forming process of Rh-110. Here, we support the transient state protection mechanism of RhB.

**Author Contributions:** Conceptualization, S.K. and S.K.R.; methodology, S.K.; software, S.K.; validation, S.K.R. and S.K.; investigation, S.K.; resources, S.K.; data curation, S.K.; writing—original draft preparation, S.K.; writing—review and editing, S.K.R.; visualization, S.K.R.; supervision, S.K.R.; project administration, S.K.R.; funding acquisition, S.K. All authors have read and agreed to the published version of the manuscript.

**Funding:** This work was financially supported by the Department of Science and Technology, Govt. of India, under the WOS-A scheme (SR/WOS-A/CS-128/2018).

**Institutional Review Board Statement:** Not applicable.

**Informed Consent Statement:** Not applicable.

**Data Availability Statement:** Not applicable.

**Acknowledgments:** We would like to show our gratitude to Central Instrumentation Facility lab of Birla Institute of Technology, Mesra for extending its support in this research work.

**Conflicts of Interest:** The authors declare no conflict of any kind of interest.

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
