# Peer review of "Enhanced Photocatalytic Oxidation of RhB and MB Using Plasmonic Performance of Ag Deposited on Bi2WO6"

_chemistry, doi:10.3390/chemistry4020022_

Round 1

Reviewer 1 Report

This work can be accepted for publication. It is well well written and well organized. Minor comments below.

Please provide additional keywords, to improve work visibility.

Please check: Fig. 6f - 101 nm?

Table 1: Measurements units should be defined as part of header.

Conclusion section should be improved.

Please check References style (in text especially).

Please check and correct text editing. There are some errors.

Author Response

We want to thank our reviewer for his valuable comments and suggestions. We have revised the manuscript according to the reviewer’s suggestions. Text and references has been thoroughly revised in the manuscript. Other comments have also been accepted and revised in the manuscript.

Reviewer 1

  • Please provide additional keywords, to improve work visibility.

Reply: Thanks to the reviewer for his suggestion. Keyword has been revised in the manuscript according to the reviewer’s suggestion.

  • Please check: Fig. 6f - 101 nm?

  Reply: Thanks to the reviewer for his comment.10 1/nm is the scale of SAED pattern.

  • Table 1: Measurements units should be defined as part of header.

  Reply: Thanks to the reviewer for his suggestion. Revision has been done according to the reviewer’s suggestion.

  • Conclusion section should be improved.

  Reply: Thanks to the reviewer for his suggestion

  • Please check References style (in text especially).

  Reply: Thanks to the reviewer for his suggestion. All the references have been revised in the manuscript.

  • Please check and correct text editing. There are some errors

  Reply: Thanks to the reviewer for his suggestion. Text has been revised in the manuscript.

Reviewer 2 Report

Rout et al. described the synthesis of plasmonic nanostructures of Bi2WO6 and Ag/Bi2WO6, and their use in photocatalytic oxidation of water pollutants (rhodamine B and methylene blue). Research on the use of various catalytic systems in photocatalytic approaches of water pollutants degradation is currently carried out by many research centers around the world. The methodology proposed by the authors is interesting and the results are presented in an understandable way. 

My critical remarks relate to the editorial aspect:

1) Throughout the manuscript, there is an inconsistency in using / not using spaces between the text and the footnotes in square brackets, as well as between values and units such as min, mL, nm, and the like. I am asking for unification.

2) An unnecessary dot (Abstract, (...)the schottky barrier..(...). Moreover, 'schottky' should be written with a capital letter - this is the last name.

3) Page 2 - no subscripts in the formula (Bi2WO6@MoS2).

4) Page 3, 2.3. Characterization, 'japan' should be capitalized.

5) Unnecessary spaces (check the whole text).

6) Page 5 - 'it' should be lowercase ('Although, It(...)).

7) Abbreviation 'Fig.' should be consistently capitalized throughout the manuscript. 

8) Page 7 - different, smaller font.

9) Low quality of Figure 14f (groups of atoms are not adjacent to bonds).

10) Low quality of equations concerning proposed mechanism (Page 21). Arrows showing the direction of the reaction should be in the center between the substrates and products.

11) Reference 1 - a lack of doi.

12) References 15, 16, 23, 24, 25, 31, 35-41, 44, 45 are incomplete (journal name and doi number missing). Please complete this missing information.

Having responded to these comments, I can recommend this manuscript for publication in Chemistry.

Author Response

I want to thank our reviewers for their valuable comments and suggestions. We have tried to revise the manuscript according to the reviewer’s suggestions. Text and references and has been thoroughly revised in the manuscript. Other comments have also been accepted and revised in the manuscript.

Reviewer 2

  • Throughout the manuscript, there is an inconsistency in using / not using spaces between the text and the footnotes in square brackets, as well as between values and units such as min, mL, nm, and the like.I am asking for unification.

    Reply: Thanks to the reviewer for his comment. The manuscript has been revised according to the reviewer’s suggestion.

2)    An unnecessary dot (Abstract, (...)the schottky barrier..(...). Moreover, 'schottky' should be written with a capital letter - this is the last name.

Reply: Thanks to the reviewer for his comment. Unnecessary dots have been removed from the text. The correction in Schottky has been done in the manuscript according to the reviewer’s suggestion.

  • Page 2 - no subscripts in the formula (Bi2WO6@MoS2).

 Reply: Thanks to the reviewer for his comment. Correction has been made in manuscript.

  • Page 3, 2.3. Characterization, 'japan' should be capitalized.

Reply: Thanks to the reviewer for his comment. Word Japan has been capitalized in the manuscript.

  • Unnecessary spaces (check the whole text).

Reply: Thanks to the reviewer for his comment. The manuscript has been rechecked according to the reviewer’s suggestion.

6) Page 5 - 'it' should be lowercase ('Although, It(...)).

Reply: Thanks to the reviewer for his comment. Correction has been made in the manuscript.

  • Abbreviation 'Fig.' should be consistently capitalized throughout the manuscript. 

     Reply: Thanks to the reviewer for his suggestion. The manuscript has been revised according to the    reviewer’s suggestion.

8) Page 7 - different, smaller font.

Reply: Thanks to the reviewer for his comment. Correction has been made in the manuscript.

  • Low quality of Figure 14f (groups of atoms are not adjacent to bonds).

   Reply: Thanks to the reviewer for his comment. High quality of Figure 14f has been inserted in the   manuscript.

10) Low quality of equations concerning proposed mechanism (Page 21). Arrows showing the direction of the reaction should be in the center between the substrates and products.

Reply: Thanks to the reviewer for his suggestion. Writing pattern of the equations has been revised in the manuscript.

11) Reference 1 - a lack of doi.

Reply: Thanks to the reviewer for his suggestion. Doi has been revised in the manuscript.

12) References 15, 16, 23, 24, 25, 31, 35-41, 44, 45 are incomplete (journal name and doi number missing). Please complete this missing information.

Reply: Thanks to the reviewer for his suggestion. All the references have been revised in the manuscript.

Reviewer 3 Report

The manuscript reports the fabrication of noble metal-semiconductor hybrid photocatalyst Ag/Bi2WO6 for the degradation of RhB and MB. The authors characterized the material thoroughly by multiple techniques and explored its photocatalytic performance. In general, relatively sufficient data were provided. However, the provided data are not that convincing to support the conclusions. In another words, the obtained photocatalyst seems not very efficient. The authors might need to consider reorganizing the data from a different perspective. Besides, the manuscript is POORLY written (errors could be easily found), the authors should carefully re-examine the manuscript to correct all the grammar mistakes or transfer the manuscript to a specialized agency for modifications. Some major considerations as follows need to be taken care of before further decision can be given.

  1. p1. Keywords should include Ag, Bi2WO6, and localized surface Plasmon resonance, replacing hydrothermal
  2. p1. The introduction part should also have a brief description of photodegradation of organic dyes since it is the aim of the present study.
  3. p2. Instead of giving some examples about the metal and semiconductor combination complex, the authors should focus more on why Ag and Bi2WO6 are selected for the present study and how this complex will improve the photodegradation of organic dyes.
  4. p7. It is contradictory to claim metal Ag from TEM while Ag2O from XPS. “beyond this temperature, the composite is stable” should be “unstable”.
  5. p12. The authors should put EIS of all the photocatalysts in one figure for better comparison.
  6. p14. The black line in the UV-vis should be Bi2WO6 instead of Ag/Bi2WO6, and I didn’t see a wider Vis-light region absorption for Ag/Bi2WO6 compared to Bi2WO6.
  7. p15. UV absorption spectra of pure RhB and Rh-110 are needed to confirm the degradation process. Mass spectra are also needed to confirm the existence of Rh-110.

Rh-110 is another organic dye, it is not an efficient degradation if RhB can only be transferred to Rh-110.

  1. p18. Figure e is quite confusing. It seems like degradation efficiency first increases and then decreases since C/C0 first increases and then decreases. The authors should make this figure clearer.
  2. p20. Does the photocatalyst fully degrade MB?
  3. p20. Other literatures about photodegradation of RhB and MB should be given as a table for comparison.
  4. p21. The mechanism part should be checked very carefully. The authors can do trapping experiments to determine the active species.

Author Response

I want to thank reviewers for their valuable comments and suggestions. We have tried to revise the manuscript according to the reviewer’s suggestions. Text and references and has been thoroughly revised in the manuscript. Other comments have also been accepted and revised in the manuscript.

Reviewer 3

  1. p1. Keywords should include Ag, Bi2WO6, and localized surface Plasmon resonance, replacing hydrothermal

Reply: Thanks to the reviewer for his suggestion. The keyword has been revised in the manuscript according to the reviewer’s suggestion.

  1. p1. The introduction part should also have a brief description of photodegradation of organic dyes since it is the aim of the present study.

Reply: Thanks to the reviewer for his suggestion. The introduction part has been revised according to the reviewer’s suggestion.

  1. p2. Instead of giving some examples about the metal and semiconductor combination complex, the authors should focus more on why Ag and Bi2WO6 are selected for the present study and how this complex will improve the photodegradation of organic dyes.

Reply: Thanks to the reviewer for his suggestion. The importance of Ag and Bi2WO6 in the present study has been added in the introduction section as per reviewer’s suggestion.

  1. p7. It is contradictory to claim metal Ag from TEM while Ag2O from XPS. “beyond this temperature, the composite is stable” should be “unstable”.

Reply: Thanks to the reviewer for pointing out this mistake. The contradictory XPS explanation about Ag2O formation has been revised in the manuscript.

The statement “beyond this temperature, the composite is stable” has been corrected and modified in the manuscript.

  1. p12. The authors should put EIS of all the photocatalysts in one figure for better comparison.

Reply: According to the reviewer’s suggestion, the separate EIS plot of all the photocatalysts in the manuscript has been replaced with a single EIS plot.

  1. p14. The black line in the UV-vis should be Bi2WO6 instead of Ag/Bi2WO6, and I didn’t see a wider Vis-light region absorption for Ag/Bi2WO6 compared to Bi2WO6.

Reply: Thanks to the reviewers for the important comment. The optical part in the manuscript has been revised according to the reviewer’s suggestion.

  1. p15. UV absorption spectra of pure RhB and Rh-110 are needed to confirm the degradation process. Mass spectra are also needed to confirm the existence of Rh-110.

Reply: Thanks to the reviewers for the comment. The mass spectra were not possible in the institute. The photocatalytic oxidation of RhB to Rh-110 is supported by the reference. The study of different breakdown products of RhB is already been reported in many literatures.

  1. p18. Figure e is quite confusing. It seems like degradation efficiency first increases and then decreases since C/C0 first increases and then decreases. The authors should make this figure clearer.

Reply: Thanks to the reviewers for the comment. Figure e shows that RhB gets degraded and then gets oxidized to Rh-110, due to which first the plot decreases and then ultimately increases. The detailed explanation of the figure has been added in the manuscript in the first paragraph of section 3.4.

  1. p20. Does the photocatalyst fully degrade MB?

Reply: The photocatalyst does not fully degrade MB. The higher MB removal efficiency of 68% in 90 min has been observed with Ag/ Bi2WO6 (1:2) Whereas Pure Bi2WO6 degrades only 38% of MB in 90 min.

  1. p20. Other literatures about photodegradation of RhB and MB should be given as a table for comparison.

Reply: According to the reviewer’s suggestion, the table for comparison of RhB and MB degradation by other photocatalysts has been added in the manuscript.

  1. p21. The mechanism part should be checked very carefully. The authors can do trapping experiments to determine the active species.

Reply: Thanks to the reviewer for the comment. Scavenger test has been performed and added in the manuscript in the last section (proposed mechanism).

Round 2

Reviewer 3 Report

The authors have addressed most of the questions and suggestions, and made careful revisions about the early manuscript. I am very pleased to see the revised manuscript, and I recommend this work for publication in Chemistry. There are a few other issues as follows that the authors can take care of to further improve the quality of the manuscript.

  1. p1. Some references are needed for the first paragraph of the introduction.
  2. p19. Figure 13 is missing. The authors might name the figures wrong and forget 13.
  3. p18 Figure e. I think the authors didn’t understand what I said by “Figure e is quite confusing. It seems like degradation efficiency first increases and then decreases since C/C0 first increases and then decreases.” Let me explain it in a different way. What do C and C0 represent here? If they are the absorbance of RhB at 554 nm, the value of C/C0 will decrease all the time even though RhB is oxidized to Rh-110.
  4. p22. The authors should explain how good their catalyst is compared to other catalysts or at least write a comparison for Table 2, instead of just putting Table 2 there.

Author Response

I want to thank reviewer for his valuable comments and suggestions. We have tried to revise the manuscript according to the reviewer’s suggestions.

  1. p1. Some references are needed for the first paragraph of the introduction.

Reply: Thanks to the reviewer for his suggestion. References have been added in the introduction section.

  1. p19. Figure 13 is missing. The authors might name the figures wrong and forget 13.

Reply: Thanks to the reviewer for pointing out this mistake. The numbering of the figures in the manuscript has been rechecked and modified.

  1. p18 Figure e. I think the authors didn’t understand what I said by “Figure e is quite confusing. It seems like degradation efficiency first increases and then decreases since C/C0 first increases and then decreases.” Let me explain it in a different way. What do C and C0 represent here? If they are the absorbance of RhB at 554 nm, the value of C/C0 will decrease all the time even though RhB is oxidized to Rh-110.
  2.  

Reply: Thanks to the reviewer for his comment. The figure e shows a decrease in the concentration of RhB between 554nm to 500nm. But at 498nm, selective oxidation takes place and RhB gets oxidized to Rh-110, the concentration of Rh-110 increases with irradiation time due to which the plot rises. Here C is the new concentration of the dye in respect to initial concentration of the dye (Co) which has been considered 1. This phenomenon is also mentioned in the literature reference no 49.

  1. p22. The authors should explain how good their catalyst is compared to other catalysts or at least write a comparison for Table 2, instead of just putting Table 2 there.

Reply: Thanks to the reviewer for his suggestion. The comparison for table 2 has been written in introduction section of the manuscript.
